# Evaluation of Ozone and its Precursors using the Multi-Scale Infrastructure for Chemistry and Aerosols Version 0 (MUSICAv0) during the Michigan-Ontario Ozone Source Experiment (MOOSE)

Noribeth Mariscal[1], Louisa K. Emmons[2], Duseong S. Jo[3], Ying Xiong[4], Laura M. Judd[5], Scott J. Janz[6], Jiajue Chai[7], and Yaoxian Huang[1]

[1]Department of Civil and Environmental Engineering, Wayne State University, Detroit, Michigan, 48202, United States
[2]Atmospheric Chemistry Observations and Modeling Laboratory, National Center for Atmospheric Research, Boulder, Colorado, 80305, United States
[3]Department of Earth Science Education, Seoul National University, Seoul, 08826, South Korea
[4]College of Environment and Climate, Institute for Environmental and Climate Research, Guangdong-Hongkong-Macau Joint Laboratory of Collaborative Innovation for Environmental Quality, Jinan University, Guangzhou, 511443, China
[5]NASA Langley Research Center, Hampton, Virginia, 23666, United States
[6]NASA Goddard Space Flight Center, Greenbelt, Maryland, 20771, United States
[7]Department of Chemistry, State University of New York College of Environmental Science and Forestry, Syracuse, New York, 13210, United States

*Correspondence to*: Yaoxian Huang (yaoxian.huang@wayne.edu)

**Abstract.** Surface ozone ($O_3$) in Southeast Michigan (SEMI) often exceeds U.S. National Ambient Air Quality Standards, posing risks to human health and agroecosystems. SEMI, a relatively small region in the state of Michigan, contains most of the state's anthropogenic emission sources and more than half of the state's population, and is also prone to long-range and transboundary pollutant transport. Here, we explore the distribution of $O_3$ and its precursors, such as nitrogen oxides ($NO_x$) and volatile organic compounds, over SEMI during the summer of 2021 using the chemistry-climate model, MUSICAv0 (Multi-Scale Infrastructure for Chemistry and Aerosols, Version 0). Using the regional refinement capabilities of MUSICAv0, we created a custom grid over the state of Michigan of 1/16˚ (~7 km) to better understand the local-scale impacts of chemical and dynamic complexity in SEMI and compared it with a grid with 1/8˚ (~14 km) resolution over the contiguous United States. Model simulations are evaluated using a comprehensive suite of observations from Phase I of the Michigan-Ontario Ozone Source Experiment (MOOSE) field campaign. MUSICAv0 with higher horizontal grid resolution showed excellent skill in capturing peak $O_3$ concentrations, but showed larger variation in the simulation of $O_3$ precursors (e.g., $NO_x$, HCHO, isoprene). In addition, we implemented a diurnal cycle for anthropogenic nitric oxide (NO) emissions, which is generally not included in global models. As a result, modeled nighttime $O_3$ was improved because of lower $NO_x$ concentrations during the night. This work shows that when conceptualizing models in urban regions, it is important to consider a combination of high horizontal

resolution and the diurnal cycle of emissions, as they can have important implications for the simulation of secondary air pollutants.

## 1 Introduction

Air pollution can significantly impact air quality (Akimoto, 2003; Fiore et al., 2002; Jacob et al., 1993), human health (Anenberg et al., 2019; Huang et al., 2020; Lelieveld et al., 2015), and climate change (Monks et al., 2015; Ramanathan et al., 2002, 2008; Unger et al., 2010). Although air quality in the United States has substantially improved since the implementation of the Clean Air Act of 1990, tropospheric ozone ($O_3$) still poses a challenge to many regions in the United States (Cooper et al., 2014, 2015; Jaffe and Ray, 2007). $O_3$, a secondary air pollutant formed through the photochemical interactions between its gas-phase precursors, nitrogen oxides ($NO_x = NO + NO_2$) and volatile organic compounds (VOCs), often exceeds allowable limits for $O_3$ established by the National Ambient Air Quality Standards (NAAQS) set by the United States Environmental Protection Agency (US EPA) (i.e., a maximum daily 8-hour average (MDA8) of 70 ppbv or less) in various US cities, despite significant reductions in its precursor species.

Southeast Michigan (SEMI) has often been classified as a nonattainment area for $O_3$ (US EPA, 2021). SEMI has experienced historically high levels of air pollution from being heavily concentrated with industry (e.g., coal-fired power plants, steel and cement facilities, petroleum refineries, and incinerators) and is subject to various mobile emissions sources due to its proximity to highways and the US-Canada ports-of-entry (in Detroit and Port Huron). Elevated $O_3$ levels have been associated with a variety of negative impacts to human health, agriculture, and the natural environment, which include premature deaths attributable to respiratory and cardiovascular diseases (Sicard et al., 2018), impacts to crop yields resulting from reduced photosynthesis (Fuhrer et al., 1997; Wortman and Lovell, 2013), and reduced visibility due to photochemical smog. The Michigan-Ontario Ozone Source Experiment (MOOSE) (Olaguer et al., 2023) was carried out to define potential $O_3$ attainment strategies in SEMI and better understand what contributes to $O_3$ exceedances in the region. It was a multi-institution (e.g., Michigan Department of Environment, Great Lakes, and Energy (EGLE), Environment Climate Change Canada (ECCC), Ontario Ministry of Environment, Conservation, and Parks (MECP), and university partners) campaign that was carried out in two phases: Phase I (24 May to 30 June 2021) and Phase II (6–28 June 2022). The MOOSE observations included a mobile lab with detailed measurements of ozone and its precursors, ground-based remote sensors (i.e., Pandora), and an airborne remote sensor (i.e., GCAS). Previous studies in Michigan have mainly investigated the impact of lake breezes on air quality (Abdi-Oskouei et al., 2020; Acdan et al., 2023; Brook et al., 2013; Dye et al., 1995; Hanna and Chang, 1995; Stanier et al., 2021; Vermeuel et al., 2019) and the connections between human health adversities and air pollution (Cassidy-Bushrow et al., 2020; Lemke et al., 2014), with little attention focused on $O_3$ atmospheric chemistry in SEMI. Xiong et al (2023) was the first to use a combination of MOOSE campaign measurements in 2021 and box modeling to investigate $O_3$ formation regimes in SEMI and found that summertime $O_3$ is limited by VOC emissions, but pointed to uncertainties due to

the small number of days used for the analysis. Because the spatial distribution of $O_3$ is dependent on precursor emissions, location, and meteorology, local $O_3$ production and loss in SEMI may be largely different compared to other regions.

Models provide credible, process-based mathematical representations of chemistry-climate interactions in the atmosphere (Brasseur and Jacob, 2017). $O_3$ biases have been identified in various global chemistry-climate models, with suggestions for improvements, such as better representation in temperature, anthropogenic emission inventories, and deposition (Schwantes et al., 2022). However, in many of these cases, the global models were not being run at horizontal and vertical resolutions fine enough to simulate ozone production and loss accurately (Schwantes et al., 2022). Although current models are efficient in reproducing rural pollutant concentrations, surface $O_3$ bias persists, which can be attributable to a coarse (>100 km) grid's ineffectiveness at reproducing urban sources and transport (Jo et al., 2013; Monks et al., 2015). The large grid cells in coarse grids artificially dilute local emissions of $O_3$ precursors, imported pollution plumes, and topography, which can alter abundance and mixing at the surface (Monks et al., 2015). There have been advancements in the use of high horizontal grid resolutions (1~28 km), which have the potential to produce more realistic simulations of $O_3$ production and loss. MUSICA (Multi-Scale Infrastructure for Chemistry and Aerosols) is a state-of-the-science unified modeling framework, allowing for seamless global and regional simulation within one model with consistent dynamics and chemistry (Pfister et al., 2020). The initial implementation of MUSICA (MUSICAv0) is a configuration of CAM-chem (the Community Atmosphere Model with chemistry) available in the Community Earth System Model Version 2 (CESM2), using the Spectral Element (SE) dynamical core, allowing for regional refinement. Several studies have taken advantage of MUSICAv0's regional refinement capabilities using custom grid applications. Schwantes et al. (2022) evaluated horizontal resolution and chemistry at varying scales (~111 km and ~14 km) over the Southeastern US and found that $O_3$ was better simulated over urban regions, particularly using the ~14 km grid and updated isoprene and terpene chemistry. Tang et al. (2022) included plume rise and a diurnal cycle of fire emissions in MUSICAv0, using the standard ~14 km resolution over the contiguous US (CONUS) and found that this addition improved MUSICAv0 simulations compared with observations. Tang et al. (2023) developed a custom grid over Africa at ~28 km in MUSICAv0 and compared it to the regional model, WRF-Chem (Weather Research Forecast with Chemistry), and found that MUSICAv0 performance was comparable to that of WRF-Chem when comparing to satellite and surface measurements of $O_3$ and carbon monoxide (CO). Jo et al. (2023) constructed two global (~112 km, ~56 km) and two regional refinement (~14 km, ~7 km) grids over South Korea for use in MUSICAv0 and found that grid resolution can heavily impact model evaluation near the surface, in particular within urban regions, as well as strongly affect the oxidation of VOCs. Lichtig et al. (2024) used a custom grid over South America with a resolution of ~28 km to quantify the local and long-range origins of CO in the region. Edwards et al. (2024) used MUSICAv0, along with the Geostationary Environment Monitoring Spectrometer (GEMS) to study $NO_x$ over Northeast Asia and Seoul, South Korea to distinguish different emission sources. As can be noted from previous work, custom grids have been used to understand an extensive range of atmospheric physical and chemical processes.

In this study, we created a custom regional refinement grid over the state of Michigan in the United States with a horizontal resolution of 1/16° (~7 km) and compared it to the standard MUSICAv0 1/8° (~14 km) grid over CONUS. We used the

Community Mesh Generation Toolkit, which is available to the community and provides the necessary tools for defining a
high-resolution grid mesh (i.e., generating input files). A sector-based diurnal cycle was applied to anthropogenic nitric oxide
(NO) emissions and was included for each resolution (Crippa et al., 2018; Jo et al., 2023). . In total, four simulations were run
during Phase I of the MOOSE campaign, which included a variety of high-resolution measurements used for model evaluation.
This work focuses on evaluating the model simulations with measurements from MOOSE, the differences between the regional
refinement grids, and changes that result from the application of the diurnal cycle for anthropogenic NO emissions.
**2 Methodology**
**2.1 Model Description**
**2.1.1 Model Overview**
Simulations with regional refinement over CONUS and Michigan were conducted using MUSICAv0. The model uses the
Spectral Element (SE) dynamical core, an unstructured grid mesh based on a cubed sphere, allowing for regional refinement
(Lauritzen et al., 2018). The standard resolution for MUSICAv0 is the uniform ne30x8 CONUS grid (hereafter referred to as
ne30x8), which is 1˚ (~111 km) over most of the globe with mesh refinement of 1/8˚ (~14 km) over CONUS and 32 vertical
layers (model top of approximately 40 km). The ne30x8 grid uses a physical/chemical time step of 225 seconds. Simulations
use the MOZART-TS2 (Model of OZone And Related chemical Tracers, troposphere-stratosphere v2) chemical mechanism,
which expands a comprehensive representation of tropospheric and stratospheric chemistry (MOZART-TS1, Emmons et al.,
2020) with more detailed gas-phase chemistry for isoprene and terpene species (Schwantes et al., 2020), aerosol microphysics
using 4-mode Modal Aerosol Module (MAM4) (Liu et al., 2016), and the simplified Volatility Basis Set (VBS) scheme (Tilmes
et al., 2019). MAM4 assists in simulating the spatial distribution of aerosols to include type, optical depth, number, and size
distributions, while the VBS scheme allows model users to better simulate secondary organic aerosols (SOA) in urban areas
through $NO_x$-dependent SOA formation (Jo et al., 2021).
Four simulations are presented in this study and were conducted from April to August of 2021, using the month of April
as a spin-up, with a particular focus on Michigan. Initial conditions for the chemical species in the simulations were generated
based on an April 2021 restart file from a 1˚ finite volume CAM-chem run using MOZART-TS1 chemistry and regridded to
the respective SE grids used here. Although the initial condition file was based on MOZART-TS1 chemistry and the additional
species in MOZART-TS2 were initiated from zero, the majority of these species are short-lived and equilibrate quickly within
the one-month spin-up period. Meteorological fields (e.g., temperature, horizontal, and vertical winds) are nudged toward
meteorological reanalysis data from MERRA-2 (Modern-Era Retrospective Analysis for Research and Applications, Version
2) (Gelaro et al., 2017) and interpolated to the resolution of the SE grids. For this study, nudging was not applied within the
state of Michigan (horizontal center of nudging window: 43˚N, 275˚W) because the original resolution of MERRA-2 is coarser
than the spectral element grids used, which could influence meteorological field calculations at finer resolutions (Jo et al.,
127 2023).

### 2.1.2 Regional Refinement over Michigan

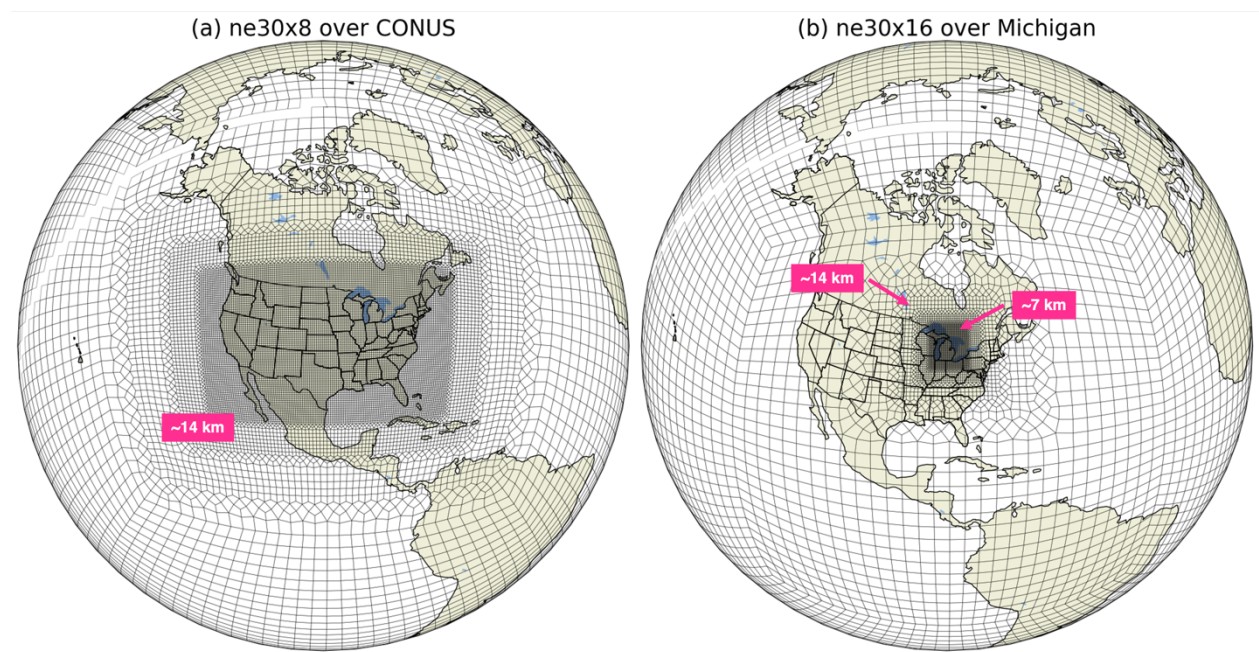

**Figure 1: The variable resolution mesh grids used for MUSICAv0 simulations in this study. (a) The ne30x8 grid and standard resolution of MUSICAv0, which is 1/8° (~14 km) over the contiguous United States (CONUS) and 1° (~100 km) for the rest of the globe, and (b) the ne30x16 grid with regional refinement over Michigan of 1/16° (~7 km), 1/8° over the majority of EPA Region 5 (includes Wisconsin, Illinois, Indiana, Ohio, and parts of Minnesota), and 1° for the rest of the globe.**

To better study the distribution of $O_3$ in SEMI, an SE grid was created over the state of Michigan using the Community
Mesh Generation Toolkit (University Corporation for Atmospheric Research, 2025). Using the Variable Resolution Mesh
Editor, the cubed sphere was rotated to have a face centered over Michigan. The 1° (ne30) base grid was further refined over
Michigan to a 1/16° (ne30x16), or approximately 7 km. The 1/16° grid then transitions into a 1/8° (~14 km) horizontal
resolution over the remainder of EPA Region 5 (includes Wisconsin, Illinois, Indiana, Ohio, and parts of Minnesota), and
finally into the 1° (~111 km) horizontal resolution over the rest of the globe. The finer resolution grid over Michigan will,
hereafter, be referred to as ne30x16. To create a smooth transition between the finer and coarser resolutions, a halo was created
around Michigan and EPA Region 5, respectively, to mitigate potential errors associated with the varying resolution changes.
The ne30x8 and ne30x16 grids are shown in **Fig. 1a** and **1b**, respectively.
MUSICAv0 simulations using the ne30x8 and ne30x16 horizontal resolutions were run with identical dynamical cores,
physics packages, and chemistry settings, but differences arise due to the different horizontal resolutions and computational
timesteps. The physics timestep specifies the number of times per model day that the physics package is called. It also defines

many other timesteps in the model through the division of the physics timestep by some integer. Scaling physics and dynamics timesteps in proportion to grid spacing is necessary for model stability. Physics and dynamic timesteps for the ne30x8 and ne30x16 grids are based on recommendations within the Community Mesh Generation Toolkit. Physics timesteps for both the ne30x8 and ne30x16 were set to 225 seconds, and dynamic time-steps at 37.5 and 18.75 seconds, respectively. The computational cost for each resolution varies based on configuration, saved output, and computational systems used. At identical model configurations, the ne30x8 and ne30x16 resolutions have computational costs of ~26,000 and ~22,000 core hours per simulated month, respectively. The finer resolution grid is about 17% more cost-efficient because it has 80,138 grid points as opposed to 174,098 in the CONUS ne30x8 grid. Configurations similar to the Michigan grid could be beneficial for local-scale studies that do not require fine resolution over an entire continent.

### 2.1.3 Emissions

MUSICAv0 has made great advances with emission dataset implementation for high horizontal grid resolutions (Schwantes et al., 2022). The model is coupled with the Community Land Model Version 5 (CLM5) (Lawrence et al., 2019), which includes the Model of Emissions of Gases and Aerosols from Nature, Version 2.1 (MEGANv2.1) algorithm to calculate biogenic emissions from vegetation (Guenther et al., 2012). Biogenic VOCs represent more than 80% of total global VOCs present in the atmosphere (Guenther et al., 1995), where isoprene alone makes up about half (Guenther et al., 2012). For this study, the specified phenology (SP) configurations of CLM are used, where MEGANv2.1 calculates biogenic emission rates in CLM based on plant functional type (PFT) distributions and leaf area index (LAI) obtained from MODIS (Moderate Resolution Imaging Spectroradiometer) (Guenther et al., 2012). Because biogenic emissions are calculated online in the model, they can vary based on horizontal resolution due to improved simulated meteorological fields (e.g., temperature) from resolving topography (Jo et al., 2023).

The anthropogenic and biomass burning emissions are conservatively regridded offline using the first-order conservative method (Jones, 1999) to the corresponding horizontal grid resolutions (i.e., ~14 km and ~7 km) used in the MUSICAv0 simulations. These regridded emissions better resolve sources and result in less artificial dilution of concentrated emissions with surrounding lower values (Schwantes et al., 2022). Emission inventory estimates are generally developed based on activity data availability for various sectors (e.g., transportation, industry, agriculture, shipping) and emission factors derived from the mass emitted per activity unit (Monks et al., 2015). Global anthropogenic emissions are from the Copernicus Atmospheric Monitoring Service Version 5.1 (CAMS-GLOB-ANTv5.1), which are monthly emissions based on EDGARv5 (Emissions Database for Global Atmospheric Research Version 5: https://edgar.jrc.ec.europa.eu/dataset_ghg50) until 2015 and then assumed until 2021 based on trends calculated from CEDSv2 (Community Emissions Data System Version 2: Hoesly et al., 2018) (Elguindi et al., 2020). CAMS-GLOB-ANTv5.1 is available at a $0.1° \times 0.1°$ spatial resolution, which is comparable to the finest resolutions of the model grids. **Table 1** shows the anthropogenic emissions of select species for SEMI in comparison to the rest of the state of Michigan to demonstrate the magnitude of SEMI emissions being represented in the model. It is important to recognize that for many of the anthropogenic emissions listed in **Table 1**, SEMI makes up about a

third of Michigan's total anthropogenic emissions. CAMS-GLOB-AIRv2.1 provides aircraft emissions from the aviation
emission inventory (Version 2.1) at a spatial resolution of $0.5° × 0.5°$ (Granier et al., 2019). Biomass burning emissions are
available through the Quick Fire Emissions Dataset (QFED) (Darmenov and da Silva, 2015) with emission factors for aerosols
and trace gases from the Fire INventory from NCAR (FINN) (Wiedinmyer et al., 2012). Other emissions, from soil, lightning,
volcanoes and oceans, are described in Emmons et al. (2020).
**Table 1: Anthropogenic emission totals for May and June 2021 based on the CAMS-GLOB-ANTv5.1 [0.1° · 0.1°] emission inventory**
**for Michigan [41.5-46°N, 230-300°W] and Southeast Michigan [41.8-43°N, 276-277.5°W].**

| Species | Molecular Weight [g/mol] | Michigan [Gg] | Southeast Michigan [Gg] |
|---|---|---|---|
| CO | 28 | 201.6 | 59 |
| NO | 30 | 34.3 | 10 |
| $SO_2$ | 64 | 19.4 | 6.5 |
| $C_2H_6$ | 30 | 1.2 | 0.3 |
| $C_3H_8$ | 44 | 1.1 | 0.5 |
| HCHO | 30 | 0.7 | 0.2 |
| BENZENE | 78 | 0.8 | 0.3 |
| TOLUENE | 92 | 3.3 | 1.6 |
| XYLENES | 106 | 6.1 | 3 |
| BIGALK* | 72 | 9.8 | 3.3 |
| BIGENE* | 56 | 1.1 | 0.4 |

*BIGALK represents lumped alkanes of C>3 (i.e., butanes, $C_4H_{10}$, and larger); BIGENE represents lumped alkenes of C>3 (i.e., butenes and larger) (Emmons et al., 2020).

**2.1.4 Application of a Diurnal Cycle for Anthropogenic Nitric Oxide Emissions**
$O_3$ has a strong diurnal variation throughout the day in the summertime, due to various processes such as precursor
emissions (i.e., $NO_x$, VOCs), solar radiation, titration by $NO_x$, dry deposition, and vertical mixing within the planetary
boundary layer (PBL) (Lin et al., 2008). $O_3$ reaches peak concentrations in the afternoon through photochemical reactions
between its precursor species in the presence of solar radiation and then decreases in the early morning through dry deposition
and $NO_x$ titration processes (Lin et al., 2008). These processes also lead to strong diurnal cycles for $NO_x$, where peak surface
concentrations are achieved in the early mornings and minimum concentrations in the afternoon. Although global models
currently account for long-range transport and emission variations, these models usually focus on concentrations of pollutants
in the daytime (Lin et al., 2008). Diurnal cycles for anthropogenic emissions are currently not considered in CESM2.

Simulating the diurnal patterns of chemical species accurately is important for assessing the impact of these atmospheric processes at maintaining this cycle (Lin et al., 2008) and are crucial factors in the evaluation of model uncertainties such as estimating long-range transport impacts on local air quality and pollution mitigation efficiency.

To better assess the present biases of $O_3$ and $NO_x$ concentrations in MUSICAv0, a diurnal cycle for anthropogenic NO emissions from CAMSv5.1 was implemented, which can strongly influence areas with high anthropogenic emissions. While emissions of other anthropogenic compounds, such as VOCs, do have diurnal variations, we have only implemented the diurnal variation for NO emissions in this work, due to its dominant role in controlling tropospheric $O_3$ and titration processes. This is based on the incorporation of the diurnal cycle presented in Jo et al (2023). NO emissions in SEMI from power generation (ENE), residential (RES), on- and off-road transportation (TNR and TRO, respectively) make up a significant amount of total NO emissions in the state of Michigan at 30, 47, 18, and 23%, respectively (see **Table S1**). In order to incorporate diurnal variations for NO emissions, we used sector- and country-specific temporal profiles based on Crippa et al (2020). The temporal profiles for each emission sector are available in the supplemental information as **Fig. S2**. Although the hourly profiles were originally developed for EDGAR, they are used in this study because both EDGAR and CAMSv5.1 emission inventories use similar sector distributions. These hourly profiles are based on the downscaling of annual emissions to hourly datasets per grid cell (Crippa et al., 2020). The diurnal cycle for anthropogenic NO emissions was applied to the ne30x8 and the ne30x16 model runs, which will, hereafter, be referred to as ne30x8 DIUR and ne30x16 DIUR, respectively.

## 2.2 Observations

### 2.2.1 Michigan-Ontario Ozone Source Experiment

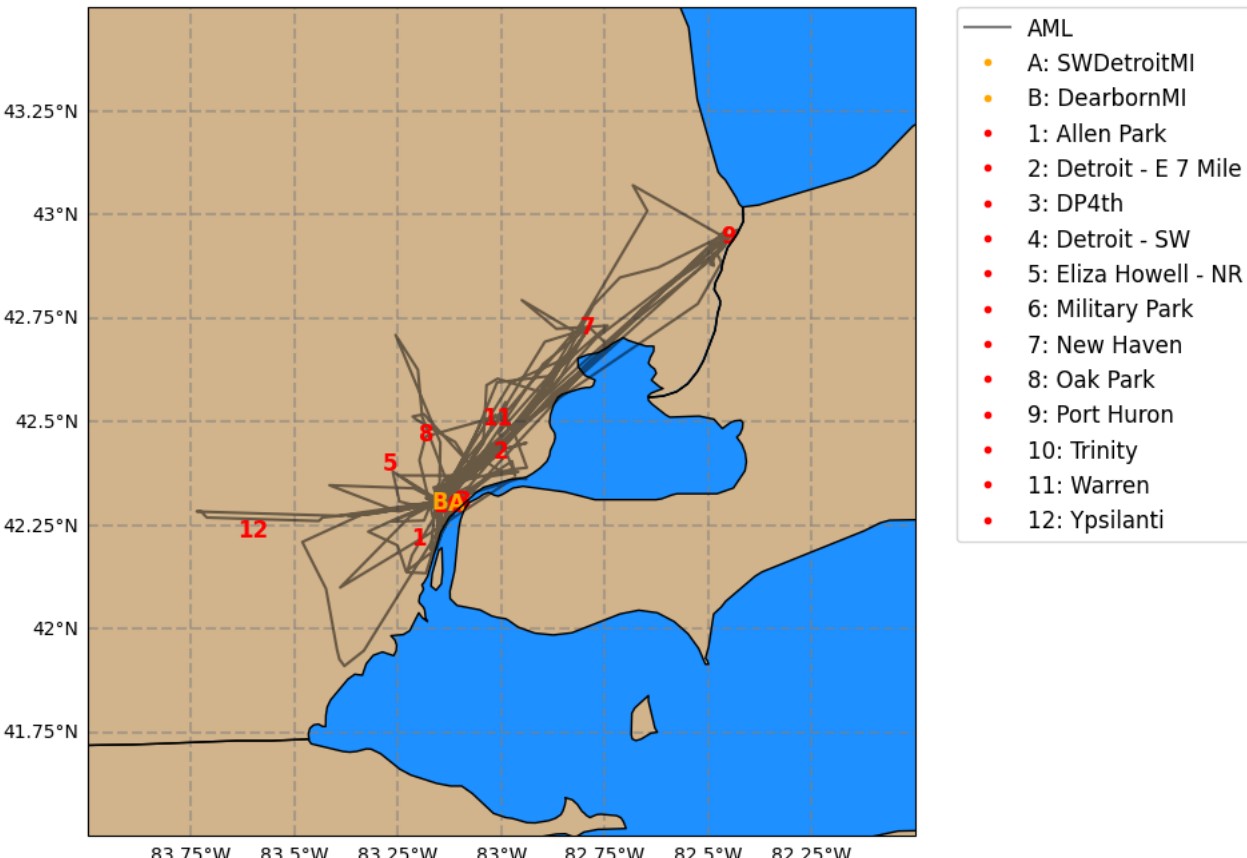

**Figure 2: Location of observations from Phase I (24 May to 30 June 2021) of the Michigan-Ontario Ozone Source Experiment (MOOSE) used in this study. The gray line shows the track of the Aerodyne Mobile Laboratory across Southeast Michigan. Stationary sites from the Michigan Department of Environment, Great Lakes, and Energy (MI EGLE) are shown as the red numbers (1-12), and the Pandora monitoring sites are shown as the yellow letters (A-B).**

With the designation of SEMI as a nonattainment area for $O_3$, MOOSE sought to determine possible attainment strategies and characterize what is driving the elevated $O_3$ levels in the region using a combination of aircraft, mobile, and in-situ measurements. The Michigan Department of Environment, Great Lakes, and Energy (EGLE) partnered with various scientific agencies including the U.S. Environmental Protection Agency (US EPA), the National Aeronautics and Space Administration (NASA), the Ontario Ministry of Environment, Conservation, and Parks (MECP), and Environment Climate Change Canada (ECCC), as well as various university partners to carry out this campaign in two phases held in the summers of 2021 (Phase I) and 2022 (Phase II), with each phase taking place for six weeks in May and June of the corresponding year. The work presented here is based on Phase I of the MOOSE field campaign. All measurement locations and tracks are shown in **Fig. 2**.

Aerodyne Mobile Laboratory (AML) measurements are available from 24 May to 30 June 2021 of Phase I. The AML drew in ambient air as it travelled throughout the SEMI region at a height of 2.8 m above ground at 8 liters per minute (Xiong et al., 2023). The Hemisphere GPS Compass (Model V100) was used to record the latitude and longitude of the AML. Consistent meteorological data of temperature, wind speed, and direction was measured by a sonic anemometer (2D RMYoung Ultrasonic Anemometer, Model 85004). The AML deployed a variety of high-resolution, real-time instrumentation, including the Vocus Proton Transfer Reaction, Time–of–Flight, Mass Spectrometer (Vocus PTR-ToF-MS), Gas Chromatograph– Electron Impact–Time–of–Flight Mass Spectrometer (GC-EI-ToF-MS), multiple Tunable Infrared Laser Direct Absorption Spectrometer (TILDAS), a LI-COR 6262 Non-Dispersive Infrared (NDIR), and a 254 nm 2BTech Model 205 Ozone Monitor. **Table 2** further elaborates on the instrumentation, types of measurements, limits of detection, and resolution. The measurements reported along the AML tracks (see **Fig. 2**) allow for further elaboration on vehicular emissions and evaporated gases. Throughout the campaign, the AML sampled ambient air continuously and remained stationary in the nighttime at the Dearborn [42.3˚N, 276.9˚W] site.

**Table 2: Detailed list of instrumentation on board the Aerodyne Mobile Laboratory during Phase I of the MOOSE field campaign.**

| Measurement | Instrument | LOD[1] | Resolution |
|---|---|---|---|
| Selected **VOCs**[1] | Vocus Proton Transfer Reaction, Time–of–Flight, Mass Spectrometer (Vocus PTR-ToF-MS) | 30-300 ppt | 1 s |
| **Speciated VOCs**[2] | Gas Chromatograph– Electron Impact- Time– of–Flight Mass Spectrometer (GC-EI-ToF-MS) | 1-20 ppt | 10 min |
| **Methane ($CH_4$) Ethane ($C_2H_6$) Formaldehyde (HCHO) Carbon Monoxide (CO) Nitric Oxide (NO) Nitrogen Dioxide ($NO_2$)** | Tunable Infrared Laser Direct Absorption Spectrometer (TILDAS) ($4^2$) | 30 ppt–3 ppb | 1 s |
| **Carbon Dioxide ($CO_2$)** | LI-COR 6262 Non-Dispersive Infrared (NDIR) | 1.5 ppb | 1 s |

| Ozone (O$_3$) | 254 nm 2BTech Model 205 Ozone Monitor | 3 ppb | 2 s |
| --- | --- | --- | --- |

[1]**Vocus PTR-ToF-MS measured for select VOCs that includes acetaldehyde, methanethiol, acrolein, acetone, furan, cyclopentadiene, isoprene, sum of MEK + butanal, benzene, sum of ethyl acetate + pyretic acid, toluene, phenol, sum of C8 aromatics, sum of C9 aromatics, sum of C10 aromatics, sum of C11 aromatics.**
[2]**GC-EI-ToF-MS measured speciated VOCs that includes aromatics, halogens, oxygenates and C$_3$-C$_{11}$ hydrocarbons.**


Vertical columns of NO$_2$ and HCHO were measured using Pandora spectrometers (Herman et al., 2009) from the Pandonia
Global Network (PGN) at two sites in SEMI – SWDetroitMI (Southwest Detroit, Michigan [42.30˚N, 276.90˚W]) and
DearbornMI (Dearborn, Michigan [42.31˚N, 276.85˚W]) – available at https://data.pandonia-global-network.org/. Pandora
uses spectroscopy to measure vertical column amounts of trace gases in the atmosphere (i.e., O$_3$, NO$_2$, HCHO), which absorb
specific wavelengths of light from the sun in the ultraviolet-visible (UV/VIS) spectrum. Pandora has the ability of retrieving
both direct-sun and all-sky radiance measurements. We use L2 direct-sun data products for NO$_2$ and HCHO columns, which
are reported to have higher precision and accuracy (Judd et al., 2019). This data product provides flags that indicate data quality
and assure usability for scientific applications (Cede, 2021; Cede et al., 2023; Liu et al., 2024). Nine data quality flags are
provided, where 0, 1, and 2 indicate assured high, medium, and low quality, respectively; data flags with a 1 in the tens position
are preliminary and not quality assured, while a 2 in the tens position is an indication of data that is unusable for science. In
this work, we applied the 0, 1, 10, and 11 data quality flags to obtain the vertical columns of NO$_2$ and HCHO per recommended
use. To obtain the tropospheric NO$_2$ column from the direct-sun product, the climatological stratospheric component for NO$_2$,
provided by PGN, was subtracted from the NO$_2$ total column. HCHO total columns were used because it is assumed that the
majority of the HCHO column can be found in the well-mixed layer (Spinei et al., 2018). The HCHO distribution was observed
within the 0-2 km altitude range and then gradually decreased with altitude, which is attributed to local surface emissions and
photochemistry near the surface (Cheng et al., 2024).
In addition to Pandora, the NASA Langley Research Center Gulfstream III (G-III) aircraft was deployed during the
campaign for 6 days between 5 June to 24 June 2021 to retrieve column density of NO$_2$ using the GeoCAPE Airborne Simulator
(GCAS) (Judd et al., 2020; Nowlan et al., 2016). GCAS is a UV/VIS spectrometer that provides NO$_2$ column measurements
(below aircraft), operating in a push-broom motion, measuring backscattered light at wavelengths between 300-490 nm
(Nowlan et al., 2018). The spatial resolution of these measurements is approximately 350 m across the track (30 pixels wide)
and 650 m along track. The sampling strategy for the G-III aircraft aims to simulate geostationary UV/VIS air quality mapping
similarly to those expected from NASA TEMPO (Tropospheric Emissions: Monitoring of Pollution) (Chance et al., 2019) by
measuring over an area of interest multiple times per day. Up to three maps per day were collected over the SEMI region
during MOOSE flight days. For this study, we use $NO_2$ columns below the aircraft from the initial release (R0), applying cloud
and glint flags.

**2.2.2 Other Observations used for Model Evaluation**

Stationary measurements throughout SEMI were used to further evaluate model simulations. of NOal-time hourly
measurements of $O_3$, $NO_2$, temperature, wind speed, and wind direction are available at various sites maintained by Michigan
EGLE, as part of the Michigan Air Sampling Network (MASN). Measurements are collected by the state of Michigan using
federal reference or equivalent monitoring methods approved by the US EPA. Data is made available at
https://www.michigan.gov/egle/about/organization/air-quality/air-monitoring. A detailed list of the sites and the observations
obtained in SEMI can be seen in **Table 3**. In addition to the sites maintained by EGLE, during the MOOSE campaign, $O_3$ and
$NO_2$ instrumentation was collocated with instruments already present at the Trinity St. Marks site in Detroit, Michigan and the
New Haven site in New Haven, Michigan.
**Table 3: List of site information and observations obtained in Southeast Michigan through the Michigan Air Sampling Network**
**(MASN) in the summer of 2021, where the numbers 1-12 are associated with Fig. 2.**

| | Site Name | Coordinates | Site Type[1] | Types of Measurements[2] |
|---|---|---|---|---|
| **1** | Allen Park | 42.22°N, 276.8°W | Suburban Downwind | $O_3$, $NO_y$, T, WS, WD |
| **2** | Detroit–E 7 Mile | 42.43°N, 277.0°W | Suburban | $O_3$, $NO_2$, T, WS, WD |
| **3** | DP4th | 42.3°N, 276.9°W | Urban | $NO_2$, T, WS, WD |
| **4** | Detroit–Southwest | 42.3°N, 276.9°W | Urban | $NO_2$, T, WS, WD |
| **5** | Eliza Howell | 42.4°N, 276.7W | Suburban, Near Highway | $NO_2$, T, WS, WD |
| **6** | Military Park | 42.3°N, 276.9°W | Urban | $NO_2$ |
| **7** | New Haven[3] | 42.73°N, 277.21°W | Rural | $O_3$, T, WS, WD |
| **8** | Oak Park | 42.47°N, 276.82°W | Suburban, Near Highway | $O_3$, T, WS, WD |
| **9** | Port Huron | 42.95°N, 277.55°W | Urban Port | $O_3$, T, WS, WD |
| **10** | Trinity St. Marks[3] | 42.3°N, 276.87°W | Urban | $O_3$, $NO_2$, WS, WD |
| **11** | Warren | 42.5°N, 277.0°W | Suburban | $O_3$ |
| **12** | Ypsilanti | 42.24°N, 276.4°W | Suburban, Near Highway | $O_3$, T, WS, WD |

[1]Description of the types of measuring locations.
[2]$O_3$ = Ozone; NO = Nitric Oxide; $NO_2$ = Nitrogen Dioxide; $NO_y$ = sum of $NO_x$ and all other reactive nitrogen; T =
Temperature; WD = Wind Direction; WS = Wind Speed.
[3]The Trinity St. Marks site contains measurements of $NO_2$, WD, and WS collected from MASN, as well as measurements of
$O_3$ and $NO_2$ from Chai et al., 2025, *In Preparation*.

## 3 Results

In this section, we evaluate MUSICAv0 model results with observations from the MOOSE field campaign in May to June
2021. For evaluation, we compare the models with $O_3$ and $NO_2$ from MI EGLE stationary sites, a range of gas-phase species
from the AML, $NO_2$ and HCHO columns from two Pandora spectrometers in SEMI, and $NO_2$ columns from GCAS. We
evaluate the models using diverse datasets from MOOSE for a comprehensive analysis, as no single dataset has the ability to
capture all aspects of atmospheric composition (e.g., emissions, chemistry, transport, meteorology). These different datasets
can also help capture different aspects of a model such as near-surface chemistry (i.e., in-situ measurements) and column
burdens (i.e., aircraft-based remote sensing), to determine model skill, characterize model errors, improve model
representation, and measure our confidence in the model results for reproducing reality.
For the comparison, we match the observed mixing ratios to the closest model grid point at each time. Modeled $NO_2$ and
HCHO columns were calculated for the troposphere using the $NO_2$ and HCHO mixing ratios at each level of the model and
multiplying it by the number density of air, which changes with altitude due to decreasing pressure, to get the number
concentrations. Once the number concentrations were obtained, we multiplied it by the layer thickness and integrated up to the
average height of the column (i.e., for Pandora, the approximate height used was ~3 km; for GCAS, the altitude of the aircraft
was ~12 km).
$O_3$ concentrations are highly associated with $NO_2$, where $NO_x$, in general, plays a critical role in the photochemical
production and destruction of $O_3$ in the presence of sunlight. $O_3$ production in the troposphere is largely dependent on the
availability of $NO_x$ and VOCs, and can give great insight on $O_3$ control. This dependency is classified into $NO_x$- and VOC-
limited regimes. In a $NO_x$-limited regime, the rate of $O_3$ production relies on the abundance of $NO_x$ and increases with $NO_x$
concentrations, but is not dependent on the concentrations of VOCs (Wang et al., 2019) . In action, decreasing $NO_x$
concentrations would lead to reductions on $O_3$ (Jacob, 1999). On the other hand, in a VOC-limited regime (or $NO_x$-saturated
regime) the rate of $O_3$ production increases with VOC concentrations and is not dependent of $NO_x$ (Wang et al., 2019), therefore
reducing the amount of VOCs would lead to reductions in $O_3$ (Jacob, 1999). The chemical relationship between $O_3$-$NO_x$-VOCs
is critically important for defining mitigation strategies set to improve $O_3$ from region to region.

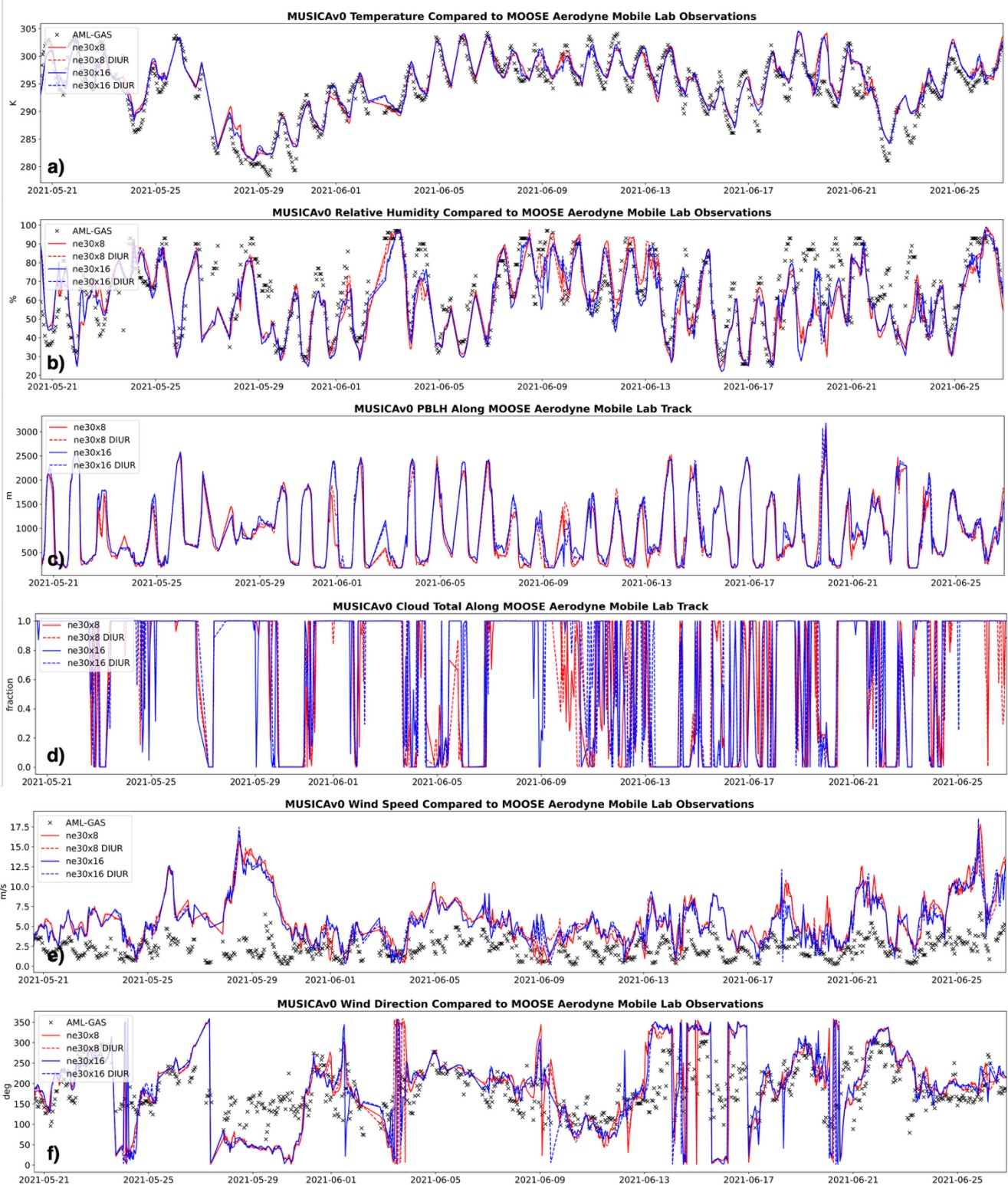


**Figure 3: Time series of (a) temperature, (b) relative humidity, (c) planetary boundary layer height, (d) cloud total, (e) wind speed, and (f) wind direction along the Aerodyne Mobile Laboratory (AML) track. Measurements of temperature and relative humidity were available and displayed as black x's in Fig. 3a and 3b. The model results are shown in red (ne30x8) and blue (ne30x16) corresponding to horizontal resolutions. The dashed lines represent model simulation results when adding the diurnal cycle for nitric oxide anthropogenic emissions, color-coded to their respective horizontal resolution.**

SEMI is a region that faces unique air quality challenges due to large industrial and automotive activity, dense population, and geographic factors. SEMI has a diverse terrain, ranging from highly urbanized areas, such as the city of Detroit, expansive agricultural lands in more remote areas, and forests surrounded by both inland and coastal lakes. The region consists of a relatively flat terrain, with a humid continental climate. Additionally, large air masses of humidity can be transported into the region from the Great Lakes (i.e., Lake Huron and Lake Erie) through the lake effect winds (Scott and Huff, 1996). A time series along the AML track of meteorological values – temperature, relative humidity, planetary boundary layer height, cloud total, wind speed, and wind direction – from the models (and observations for temperature and relative humidity) are shown in **Fig. 3**. During the campaign period in the summer of 2021, temperatures reached up to approximately 305 K and relative humidity to almost 100%. The planetary boundary layer reached more than 2500 m on most days, while cloud total was relatively varied. Modeled wind speeds follow the trend for the campaign period quite well, but are comparatively high compared to the observations, while wind directions perform generally well except on some specific days. The AML track covered a large part of the SEMI region, making its way through both very urban and rural areas. Meteorological parameters, such as temperature, are highly impacted by urbanization through the reductions in vegetated land cover and increases in energy consumption (Wang et al., 2021). Urbanization can lead to higher temperatures, and thus increasing $O_3$ production. In the simulations presented here, meteorological parameters (i.e., temperature and horizontal winds) are nudged towards reanalysis data to obtain a more realistic depiction of reality in the coarser resolution regions, leaving the regional refinement area to freely run, as the resolution of the refined area is finer than the resolution of the reanalysis dataset that is being used. Regional refinement grids, with high horizontal resolution, are capable of resolving areas with large geographical differences (Jo et al., 2023). Meteorological fields in these simulations are generally consistent indicating that meteorology is performing similarly, even with the changes in horizontal resolution. Although temperature, relative humidity, and planetary boundary layer height remain consistent among all the simulations, cloud total varies between the simulations, which can significantly impact photochemical production.

## 3.2 Evaluation with Stationary Sites

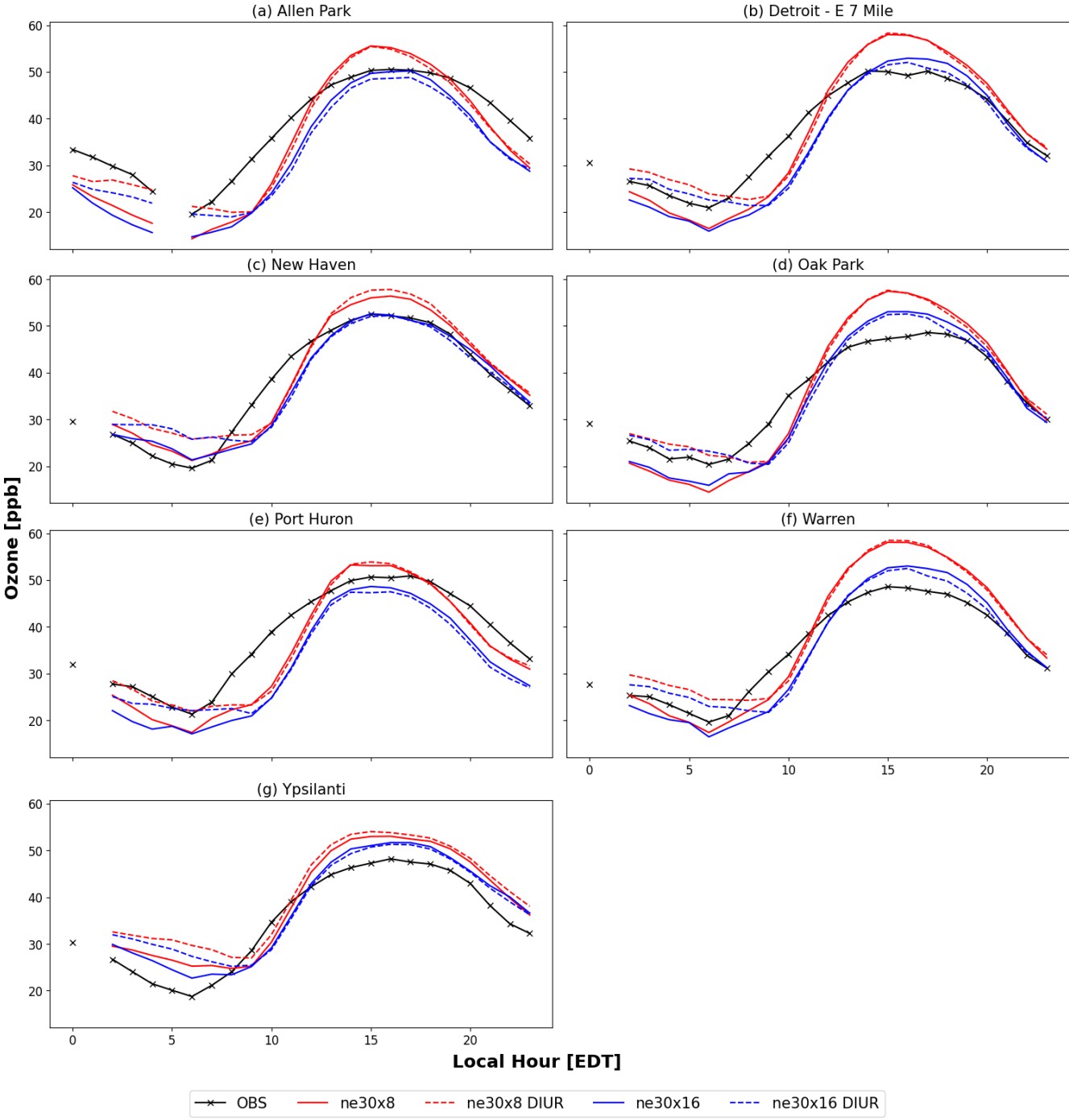

**Figure 4: Model evaluation of hourly averaged diurnal profiles of ozone concentrations at the surface during the Michigan-Ontario Ozone Source Experiment (24 May to 30 June 2021) at seven stationary measurement sites in Southeast Michigan – a) Allen Park [42.2°N, 276.8°W]; b) Detroit – E 7 Mile [42.4°N, 277.0°W]; c) New Haven [42.7°N, 277.2°W]; d) Oak Park [42.5°N, 276.8°W]; e) Port**

In this section, we evaluate the model results from four simulations during Phase I of the MOOSE field campaign (24 May to 30 June 2021) with real-time hourly measurements of $O_3$ and $NO_2$ from available stationary sites in SEMI, maintained by Michigan EGLE, as part of MASN. The stationary sites are located in an environment with mixed urban, suburban, and rural plumes (**Figure 2**; descriptions in **Table 3**). For $NO_2$, available measurements are primarily located in urban and suburban areas.

The evaluation of the four model simulations with stationary measurements for $O_3$ at seven locations in SEMI – Allen Park (Suburban Downwind), Detroit-E 7 Mile (Suburban), New Haven (Rural), Oak Park (Suburban, Near Highway), Port Huron (Urban Port), Warren (Suburban), and Ypsilanti (Suburban, Near Highway) – are shown in **Fig. 4** as a time series of their hourly averaged diurnal profiles during the MOOSE campaign. **Table S2** lists the mean biases (MB), root-mean squared error (RMSE), and Pearson correlation (CORR) for $O_3$ at the selected stationary sites. In general, the ne30x16 simulations without diurnal cycle implementation performed well compared to stationary observations with overall mean biases of -0.85, -1.12, -0.52, and 2.46 ppb for the New Haven, Oak Park, Warren, and Ypsilanti sites, respectively. Adding the diurnal cycle for NO further improved mean biases at the New Haven, Oak Park, and Warren sites with overall mean biases of -0.22, 0.02, and 0.47 ppb, respectively. During the 9-11 EDT, all model simulations miss the mark at all sites when the slope increases in the morning, which coincides with higher modeled $NO_2$ concentrations (see **Fig. 5**). On the other hand, **Fig. 4** shows that ne30x8, with and without diurnal cycle implementation, tends to overestimate $O_3$ concentrations during peak ozone times (12-18 EDT) with mean biases of up to 10 ppb. During this timeframe, the changes from the addition of the diurnal cycle for NO are minimal, with the largest differences resulting from the changing horizontal resolution. Increasing horizontal resolution reduces $O_3$ concentrations, bringing them closer to the observational datasets. A larger difference in modeled $O_3$ compared to observations occurs during minimum $O_3$ times (3-9 EDT), where differences can exceed 12 ppb. During these times, $O_3$ concentrations tend to be underestimated at most sites with the exception of New Haven and Ypsilanti, where $O_3$ concentrations from the models are higher than the observations. The application of the diurnal cycle for anthropogenic NO emissions in both horizontal resolutions showed overall improvements in $O_3$, likely as a result of better performance in $NO_2$, which is demonstrated in **Fig. 5**.

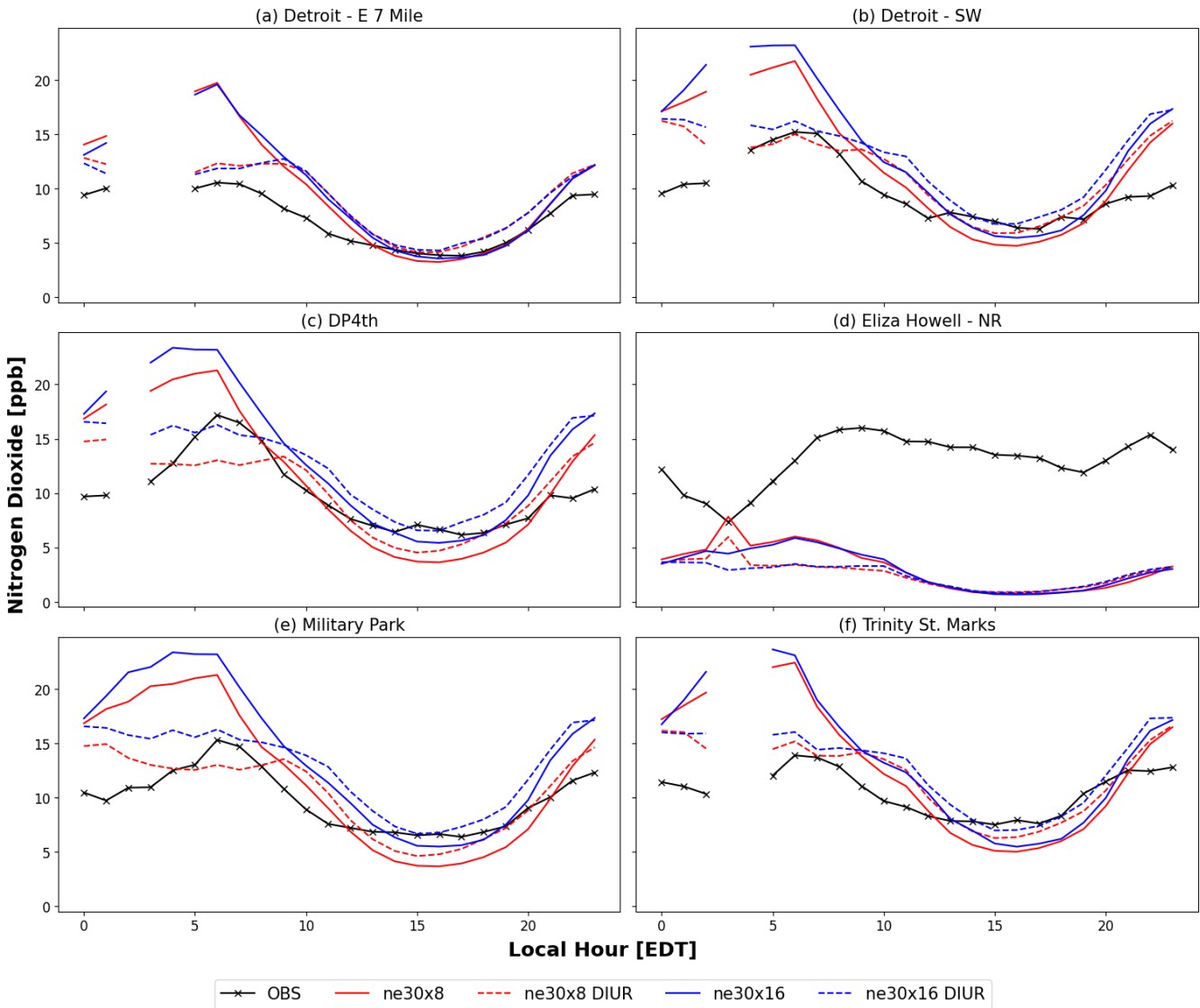

**Figure 5: Same as Fig. 4, but for NO₂ with select stationary measurements from a) Detroit – E 7 Mile [42.4°N, 277.0°W]; b) Detroit – SW [42.3°N, 276.9°W]; c) DP4th [42.3°N, 276.9°W]; d) Eliza Howell – NR [42.4°N, 276.7°W]; e) Military Park [42.3°N, 276.9°W]; and f) Trinity St. Marks [42.3°N, 276.9°W].**

**Figure 5** shows the hourly averaged diurnal profile of NO₂ at six locations in SEMI – Detroit-E 7 Mile (Suburban), Detroit-SW (Urban), DP4th (Urban), Eliza Howell-NR (Suburban), Military Park (Urban), Trinity St. Marks (Urban). **Table S3** list the statistical data for NO₂ concentrations at these stationary sites. NO₂ concentrations in the default simulations at the ne30x8 and ne30x16 horizontal resolutions had mean biases between 2-4 ppb. After implementing the diurnal cycle for NO anthropogenic emissions mean biases shifted between 0-3 ppb (dashed line in **Fig. 5**), with the exception of the Eliza Howell–NR site, which was greatly underestimated in all model simulations with overall absolute mean biases of up to 11 ppb. This large underestimation at the Eliza Howell–NR site is highly attributable to the near-road transportation emissions that were

379 not captured by the model. In general, although there are differences in NO₂ concentrations from the changing resolutions,
where urban sites showed an increase in concentrations when going to finer resolution, the large differences came from the
addition of the NO diurnal cycle. During peak times, the default configurations at each resolution showed higher concentrations
of NO₂. Adding the diurnal cycle lowered these concentrations bringing them closer to the observations. The application of
the diurnal cycle for NO lowers NO emissions in the nighttime, bringing concentrations closer to the observed values during
those times, which could in turn affect O₃ concentrations.

**3.3. Evaluation with Aerodyne Mobile Laboratory**

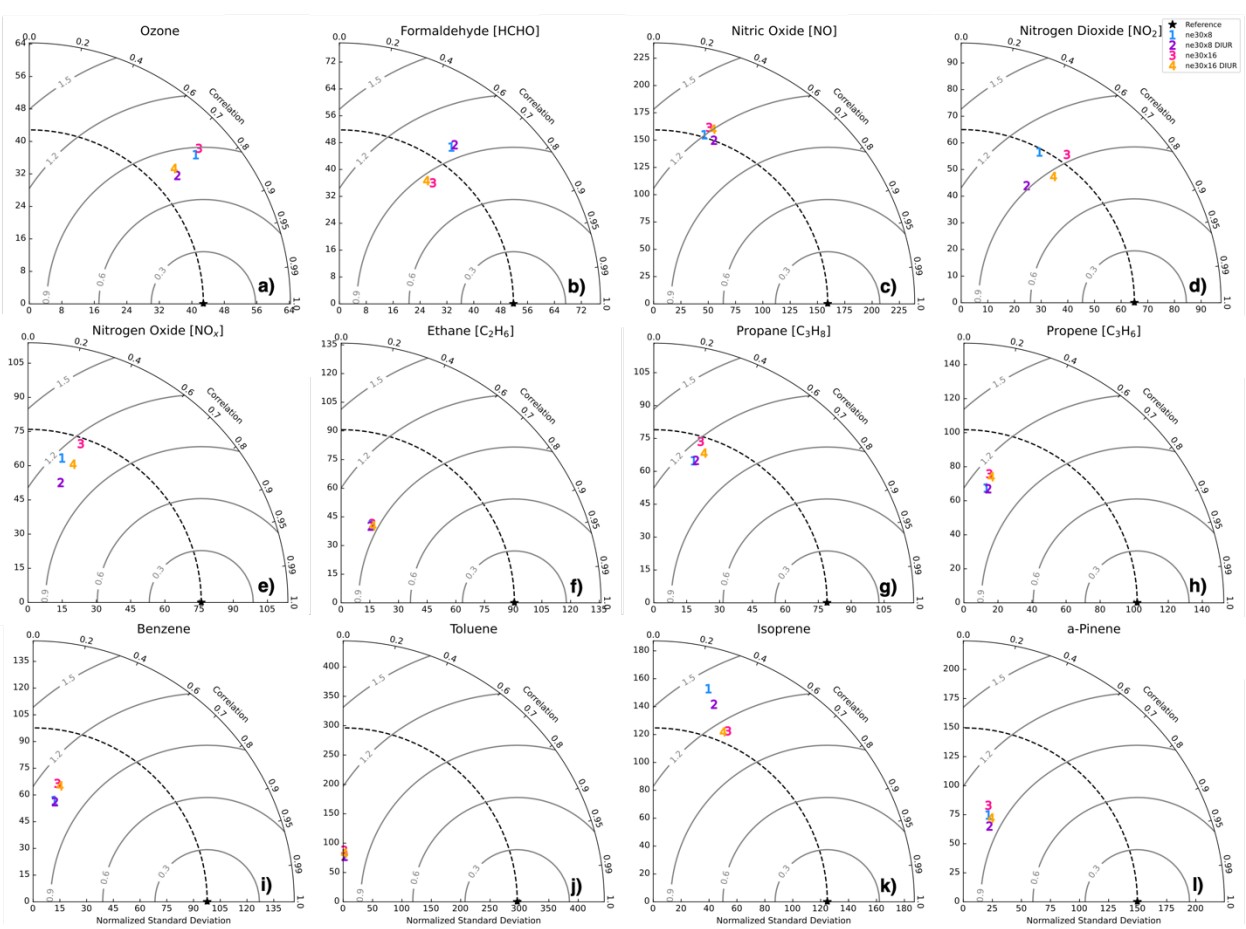

**Figure 6: Taylor diagrams comparing gas-phase species from the Aerodyne Mobile Lab (AML) to MUSICAv0 simulations during the Michigan-Ontario Ozone Source Experiment (24 May to 30 June 2021). The different simulations are represented by the numbers, where each color represents a different model configuration. The reference point (star symbol) represents the observations from the AML. The correlation corresponds to the angular axis and the normalized root-mean squared error to the radial axis.**

Here, we evaluate the four MUSICAv0 simulations against mobile observations obtained from AML during Phase I of
the MOOSE field campaign using the Taylor diagram (Taylor, 2001). Taylor diagrams allow us to summarize how closely
model simulations match with observations using a combination of the Pearson correlation, the centered normalized root-

mean-square difference, and the normalized standard deviation. A quality assurance flag was applied to the AML dataset, where we filtered the data to exclude measurements affected by traffic or self-sampling. **Figure 6** compares gas-phase species from the AML to four MUSICAv0 simulations, where the observations from AML are used as the reference (black star). Detailed statistics for all available gas-phase species and meteorological parameters can be found in **Table S4** and **S5**, respectively. The further the simulation results are from this reference point indicates poorer model performance. Of the chemical species presented in **Fig. 6**, differences based on the different regional refinement grids were significant for some species (i.e., HCHO, isoprene), while for others the differences were more dependent on the application of the diurnal cycle (i.e., $O_3$). These differences are discussed in Sect. 4. For surface $O_3$ concentrations (**Fig. 6a**), applying a diurnal cycle for anthropogenic NO increased performance in both regional refinement grids, with little difference between the grid resolutions. In contrast, NO, $NO_2$, and $NO_x$ (**Figs. 6c, 6d, 6e**) performance varied compared to AML measurements, where NO concentrations at the surface performed similarly between all model configurations, with slight improvements when adding the diurnal cycle. $NO_2$ and $NO_x$ simulation, on the other hand, did see a larger impact with both grid resolution and diurnal cycle application, where the ne30x16 run performed best compared to observations. The differences in grid resolution are seen more strongly than the inclusion of diurnal NO emissions for HCHO concentrations in **Fig. 6b**. Isoprene is the main precursor of HCHO at the surface and a stimulant of $O_3$ production (Wolfe et al., 2016). **Figure 6k** shows that isoprene is simulated better with the ne30x16 grids, which can be associated to grid resolution. Grid resolution can have a more significant impact on isoprene because BVOCs are calculated online in the model, where spatial resolution can impact meteorological fields and affect BVOC calculations. Although temperatures are not greatly affected by grid resolution, as was seen in **Fig. 3**, cloud totals are different in the two resolutions , which can impact the amount of solar radiation reaching the surface Clouds in the model can be impacted by several changes, such as changes in aerosols, which is out of the scope of this study, or related to changes in meteorology (e.g., winds). Yan et al. (2023) demonstrated that aerosols are able to impact precursor accumulation and photolysis (e.g., isoprene), where tropospheric chemical loss is enhanced due to photolysis and $NO_x$ accumulation. Cheng et al. (2022) also found that changing clouds in chemical transport models can impact photochemical reaction rates and BVOCs. Future work on evaluating model grid resolution and diurnal cycle impacts on $O_3$ formation should look more closely into aerosol-cloud interactions and how they impact photochemical production in SEMI. Additionally, **Fig. 6l** shows α-pinene, which performs similarly across all simulations, regardless of the resolution and application of a diurnal cycle for anthropogenic NO emissions. Unlike isoprene, where changes from the BVOCs calculations in MEGANv2.1 were more pronounced, α-pinene was generally unchanged across all of the simulations. **Figures 6f-6h** includes ethane ($C_2H_6$), propane ($C_3H_8$), and propene ($C_3H_6$), respectively, which are important hydrocarbon precursors of $O_3$ in areas with many anthropogenic sources. $C_2H_6$ is primarily emitted via extraction and processing of fossil fuels, while $C_3H_8$ and $C_3H_6$ are emitted mainly through petroleum gas industries (Emmons et al., 2020). All MUSICAv0 simulations generally perform similarly when compared with the observations of these species, with relatively low correlation. The models underestimate $C_2H_6$, $C_3H_8$, and $C_3H_6$, which is an indication of missing emission sources. Benzene and toluene are also important $O_3$ precursors, emitted from anthropogenic sources (Fang et al., 2016) such as solvent usage, incomplete combustion, industrial coatings, and the petroleum

industry. Both benzene and toluene (**Figs 6i, 6j**) are underestimated in the models, again, likely as a result of missing emission sources, with a small improvement in benzene as a result of grid resolution. Percent differences are shown in **Fig. S1** in the supplemental information to further demonstrate the consistent misrepresentation of $C_2H_6$, $C_3H_8$, $C_3H_6$, benzene, toluene, and α-pinene in MUSICAv0 regardless of model modifications. As chemistry-climate models move to finer resolutions (<10 km), local emission sources will need to be represented more accurately for proper use in fine-scale scientific applications. Additionally, future work should focus on evaluating simulations with the application of diurnal cycles to all anthropogenic emission sources, as they can vary greatly during the day from sector to sector.

**3.4 Evaluation with Pandora Spectrometers**

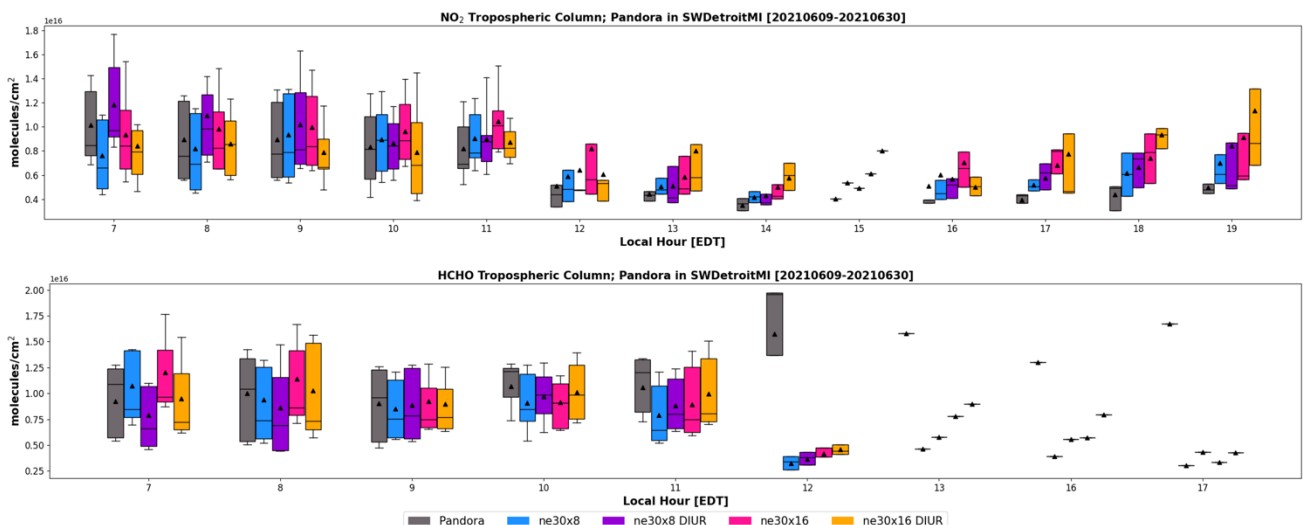

**Figure 7: Hourly binned box-and-whisker plots showing Pandora NO₂ (top) and HCHO (bottom) tropospheric columns (in grey) and modeled tropospheric columns at the SWDetroitMI [42.30°N, 276.90°W] site in Southeast Michigan. The tropospheric columns from the model simulations were calculated up to approximately 3.3 km (10 model levels) as the average height obtained from the Pandora Spectrometers was about 3 km. The box-and-whisker plots show the 10, 25, 50, 75, and 90th percentiles, where the triangles are representative of the means.**

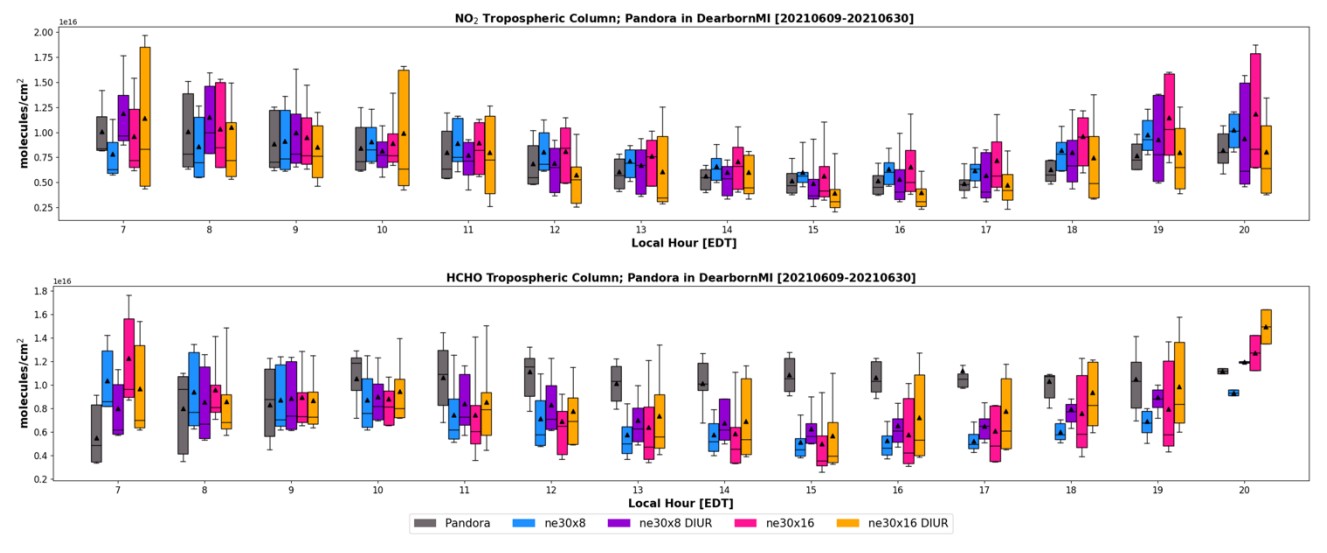

442

**Figure 8: Same as Fig. 7, but for the Pandora spectrometer located at the DearbornMI [42.31˚N, 276.85˚W] site in Southeast Michigan.**

We compare $NO_2$ and HCHO tropospheric columns from two Pandora spectrometers to the four MUSICAv0 simulations. Both Pandora monitoring sites (SWDetroitMI and DearbornMI) were located in an industrial and high-traffic setting, providing continuous observations in urban conditions and complementing the other observations. **Figures 7 and 8** show hourly binned box-and-whisker plots of $NO_2$ and HCHO columns between 9 June and 30 June 2021 for the SWDetroitMI and DearbornMI sites in SEMI, respectively, compared to the model simulations. These locations are presented in **Fig. 2**. All of the model simulations performed well when compared to $NO_2$ columns from Pandora. Overall, observed means were lower than modeled means, indicating an overestimation in the modeled $NO_2$ column. During peak $NO_2$ timeframes (7-11 EDT) at the SWDetroitMI site, modeled $NO_2$ columns saw improvements resulting from the application of the diurnal cycle for anthropogenic NO. In the afternoon, $NO_2$ columns in the models gradually increased going from coarser to finer resolution and with the added diurnal cycle. Although the difference in grid resolution plays a role in the simulation of $NO_2$ columns, the differences were not as significant in the later part of the day. The modeled and observed $NO_2$ columns were better represented when both higher resolution and the application of the diurnal cycle for anthropogenic NO were included with correlations of 0.28 and 0.31 at the SWDetroitMI site and 0.61 and 0.58 at the DearbornMI site. Consequently, HCHO columns also performed well at the SWDetroitMI and DearbornMI during the early morning (7-9 EDT). On the other hand, after 10 EDT, the models begin underestimating HCHO columns with differences of nearly a factor of 2. The locations of the Pandora spectrometers are in highly industrialized, urban areas. The large model bias in HCHO columns could be an indication of missing emission sources in the area. In addition, as was mentioned in Sec. 3.3, cloud formation also changes in the simulations, which could impact BVOC emissions, such as isoprene (main precursor of HCHO) and photolysis rates, and ultimately impact HCHO columns. A combination of grid resolution and the diurnal cycle is largely responsible for increased HCHO columns, bringing

the model closer to the measurements with correlations for the ne30x8 and ne30x16 model runs of 0.30 and 0.22 at the
SWDetroitMI site, and 0.38 and 0.30 at the DearbornMI site, respectively. Detailed statistics for $NO_2$ and HCHO tropospheric
columns from Pandora at the SWDetroitMI and Dearborn MI sites, along with the four model simulations can be found in
**Table S6** and **S7**, respectively.
**3.5 Evaluation with GCAS**

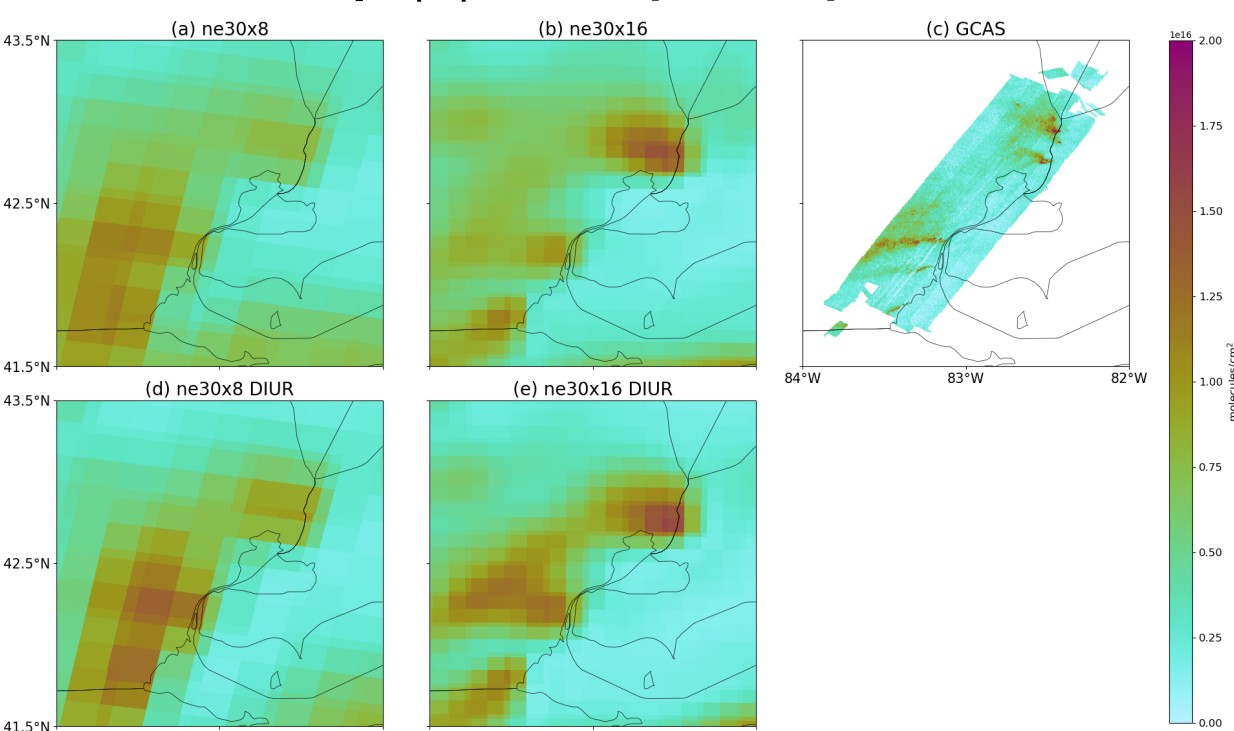


**Figure 9: Modeled and observed $NO_2$ tropospheric columns over Southeast Michigan on 11 June 2021. The GCAS instrument flew**
**over the Southeast Michigan region between 10:10 and 11:45 EDT, so modeled $NO_2$ tropospheric columns were calculated using the**
**11 EDT time frame. Figures 9a, 9b, 9d, and 9e represent modeled $NO_2$ tropospheric columns calculated to about 12 km in altitude,**
**which was the average flight altitude of the NASA Gulfstream-III aircraft. Figure 9c shows observed $NO_2$ tropospheric column from**
**the GCAS instrument during the morning time.**

**NO$_2$ Tropospheric Column [20210611 R2]**

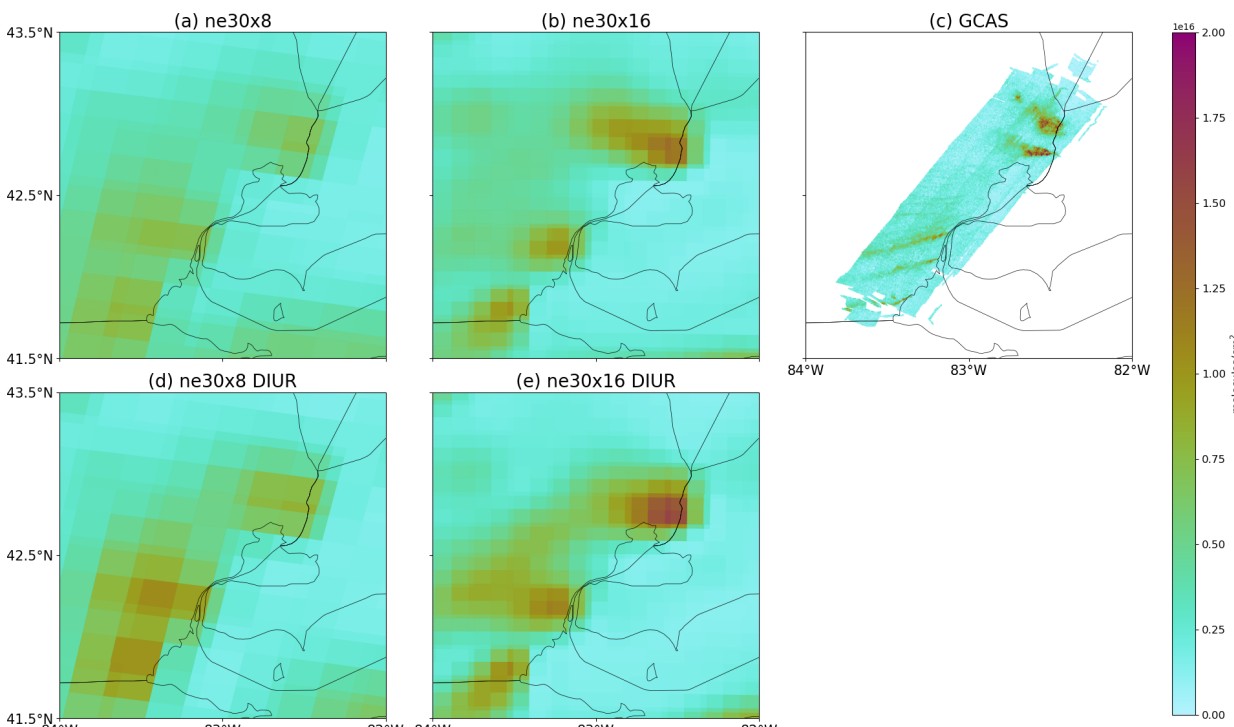


**Figure 10: Same as Fig. 9, but the GCAS instrument flew over Southeast Michigan from 11:45 to 13:16 EDT, and modeled NO$_2$**
**tropospheric columns were calculated during the 12 EDT time frame.**

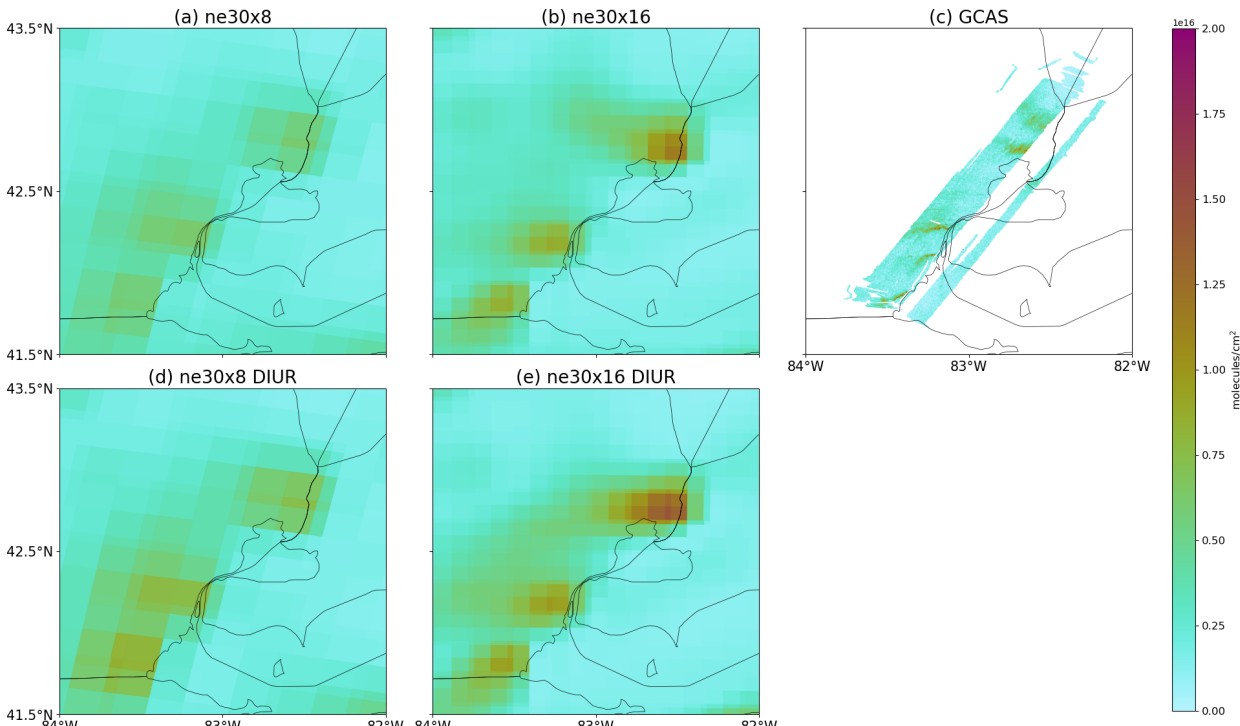

**NO$_2$ Tropospheric Column [20210611 R3]**

(a) ne30x8  (b) ne30x16  (c) GCAS

(d) ne30x8 DIUR  (e) ne30x16 DIUR


**Figure 11: Same as Fig. 9, but the GCAS instrument flew over Southeast Michigan from 13:16 and 14:00 EDT, and modeled NO$_2$**
**tropospheric columns were calculated during the 13 EDT time frame.**
In this section, we qualitatively evaluate modeled NO$_2$ tropospheric columns against observed NO$_2$ tropospheric columns
from the GCAS instrument onboard the NASA G-III research aircraft. While GCAS measures the column amount of NO$_2$
below the aircraft, the surface concentrations generally dominate the column in the lowest part of the atmosphere. **Figures 9-**
**11** show the comparison of the modeled NO$_2$ columns from the four simulations discussed in this paper with the observed
GCAS NO$_2$ columns in SEMI on 11 June 2021 for the three flights of the day. The details for each flight day can be found in
**Table S8**, and day-to-day variabilities for the remaining days can be found in the **Supplemental Information, Figs. S3-S15**.
The day of Friday, 11 June 2021 exhibited a moderate air quality index (AQI) with temperatures between 24 and 30˚C and
calm wind speeds at 2-5 m/s. The overall wind direction during the flight times was blowing from the east direction in SEMI.
In the area, plumes of NO$_2$ can be observed from source locations such as power plant emissions in Monroe, Michigan, mobile
and industrial emissions in Detroit, Michigan, additional power generation emissions in East China, Michigan, as well as
emissions from Sarnia's "Chemical Valley" in Ontario, Canada, which includes various petrochemical facilities.
**Figures 9-11** demonstrate the hourly variabilities of NO$_2$ columns in the model and observations for 11 June 2021. Three
rasters were sampled on this day, between mid-morning and mid-afternoon. In general, NO$_2$ tropospheric columns from GCAS
were higher in the morning than they were in the afternoon. All four model simulations followed a similar trend (in **Figs. 9-11**
and in the **SI**), where NO$_2$ columns were higher in the mornings compared to the afternoon. Although NO$_2$ source regions are

identifiable in all of the model simulations, the finer grid mesh better resolves the source regions and makes $NO_2$ plumes more visible in all of the time frames. The model simulations at the ne30x16 resolution (**Figs. 9-11b, 9-11e**) show good agreement with the observed wind direction blowing from the northeast direction pushing $NO_2$ in the western direction (noted in **Table S8**). In general, the magnitude differences between the coarse and fine grids, and the application of the diurnal cycle for anthropogenic NO did not have a large impact on $NO_2$ columns between simulations. What can be noted, is that the ne30x16 horizontal resolution showed more pronounced pollution plumes from source regions and more defined $NO_2$ tropospheric columns. The direction of the pollution plumes are supported by plots of temperature and wind vectors in Figs. S16-S31 in the SI for each of the flight days. Even with a resolution of 1/16˚ (ne30x16), some point sources captured by GCAS are not captured by the model because it is still relatively coarse for urban applications. With the future release of MUSICAv1, which uses the non-hydrostatic dynamical core MPAS (Model for Prediction Across Scales; on an unstructured grid mesh based on centroidal Voronoi tessellations (Du et al., 1999)), allowing for regional refinement below 5 km, estimates of emissions at finer scales over regions of interest are necessary. Tropospheric $NO_2$ columns from satellites (e.g., TROPOMI, OMI) have been used to estimate $NO_x$ emissions in localized environments (Goldberg et al., 2024; Martínez-Alonso et al. 2023; Dix et al., 2022; Beirle et al., 2019; Liu et al., 2016). For example, Martínez-Alonso et al. (2023) used TROPOMI $NO_2$ columns to derive emissions from mining and industrial activities in the Democratic Republic of Congo and Zambia and Goldberg et al. (2024) used a combination of aircraft remote sensing (i.e., GCAS), source apportionment models, and regression models to investigate $NO_2$ emissions from individual sources in Houston, Texas. Future work should take into consideration the use of the GCAS observations to develop emission inventories for use in multi-scale model simulations of Michigan.

Section 3 has evaluated the model simulations against four different types of observations obtained during MOOSE 2021. Taken together, the model evaluation shows (i) that refining the horizontal grid resolution in the model is the dominant factor leading to reductions in bias for peak $O_3$ concentrations, enhances $NO_2$ source region plumes, and better separates contrast between urban and suburban locations, such as Allen Park and Trinity St. Marks; (ii) that the diurnal cycle for anthropogenic NO emissions corrects the early morning biases in $NO_2$ and slightly impacts $O_3$, while having small impacts on peak $O_3$ values; and (iii) the high biases in VOCs points to deficiencies in the emission inventory rather than grid resolution and temporal allocation. These findings motivate the more in-depth analysis described in Sec. 4, where we discuss resolution- and diurnal emission-driven changes governing $O_3$ production and loss across SEMI.

**4 Discussion: Impacts of Grid Resolution and Diurnally Varying Emissions**

The previous section (Sect. 3) evaluated four MUSICAv0 simulations using two different regional refinement grids (ne30x8, ne30x16) and the addition of a diurnal cycle for anthropogenic NO emissions (ne30x8 DIUR, ne30x16 DIUR) against observations from Phase I of the MOOSE field campaign. Building on the evaluation in Sec. 3, this section discusses the differences resulting from changes in horizontal grid resolution and the application of the diurnal cycle for anthropogenic NO. We analyze how site-specific behaviors are driven by the model changes, and how those behaviors drive $O_3$ formation. First,

the spatial distributions in the two resolutions are compared (Figures 12-14), and then the diurnal cycles in different
environments are compared (Figures 15-18).

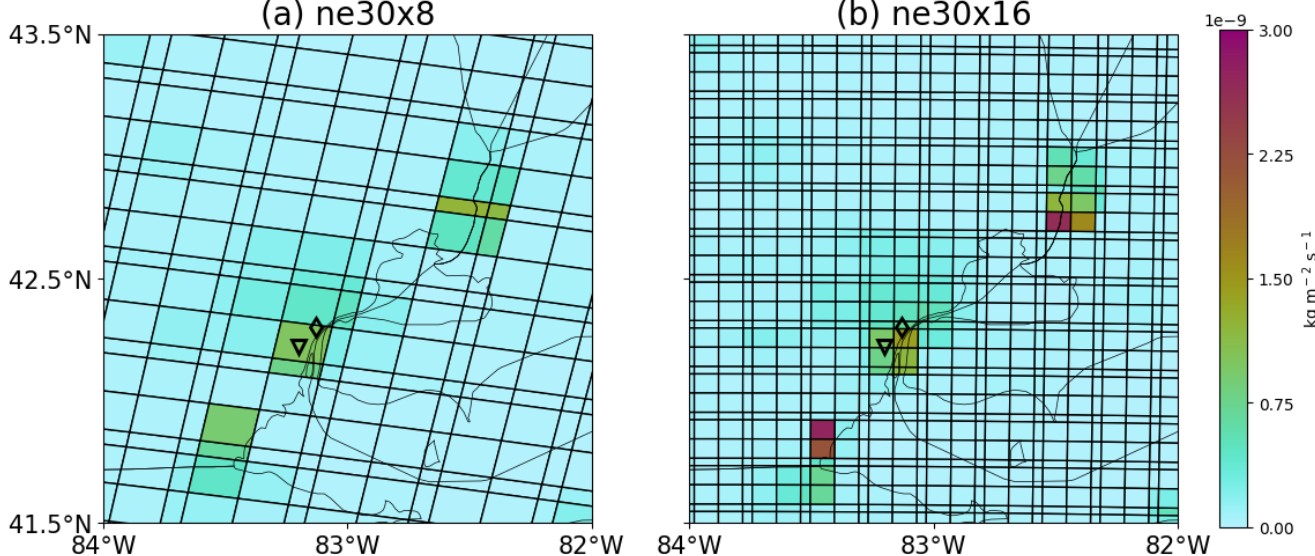


**Figure 12: Nitric oxide (NO) emission distribution averaged for June and July 2021 over Southeast Michigan on corresponding**
**regional refinement grids. Two SEMI sites – Allen Park (triangle) and Trinity St. Marks (diamond) – are shown relevant to their**
**grid box locations.**
**Figure 12** shows emissions of NO across the SEMI region and the different grid boxes pertaining to the (a) ne30x8 and
(b) ne30x16 horizontal resolutions. The Allen Park and Trinity St. Marks ground sites (black triangle and diamond,
respectively) are also shown relative to their grid box locations and the distribution of NO emissions. This figure shows that
in the coarse resolution (**Fig. 12a**), the two sites (a suburban and an urban site) are represented by the same grid box, whereas
in the finer, ne30x16 resolution (**Fig. 12b**), they are present in distinct grid boxes. Although the total emissions for a region
are the same, emission fluxes can become more resolved moving to a finer grid resolution (Jo et al., 2023), which will
ultimately impact model simulation evaluation as horizontal resolution becomes finer and finer.

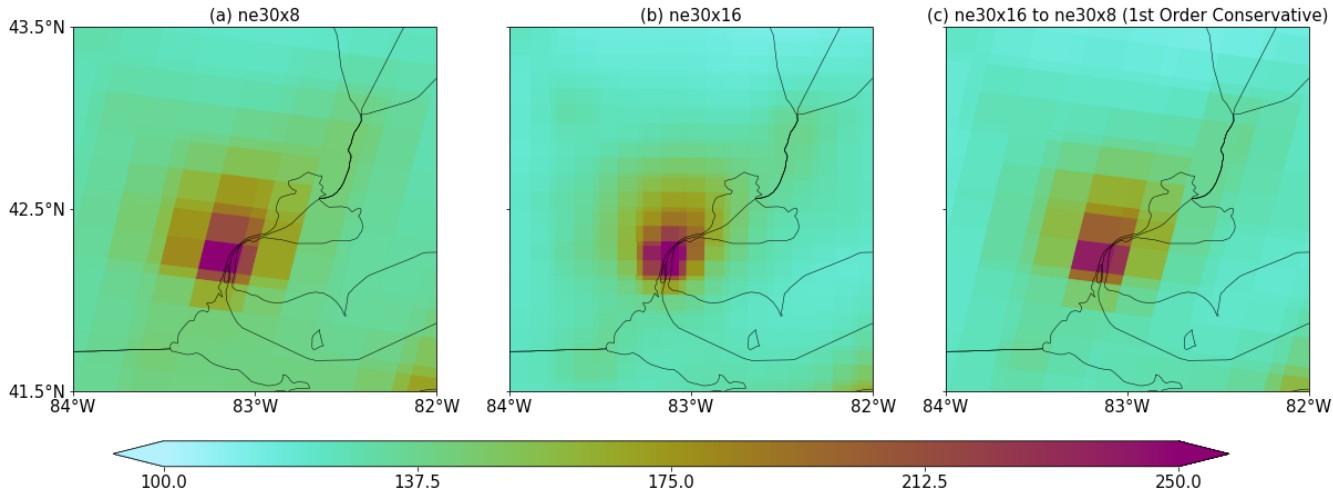


Figure 13: Modeled carbon monoxide (CO) concentrations at the surface for June 2021, where (a) is the ne30x8 horizontal resolution, (b) is the ne30x16 horizontal resolution, and (c) is the ne30x16 model output regridded to the coarser ne30x8 horizontal resolution using the first order conservative method.

To quantitatively assess the impact of the finer resolution on the simulation of ozone and its precursors, the ne30x16 (7 km) results have been conservatively regridded to the ne30x8 grid. These regridded results illustrate the impact model resolution can have on atmospheric chemistry, depending on the compound. **Figure 13** shows the modeled monthly averaged carbon monoxide (CO) concentrations at the surface for June 2021 for the ne30x8 (Fig. 13a), ne30x16 (Fig. 13b), as well as the conservatively regridded model output from the ne30x16 simulations to the ne30x8 horizontal resolutions (Fig. 13c), respectively. CO is mainly emitted through incomplete combustion processes and has a generally long lifetime, lasting from week to months, allowing it to be transported over long distances (Gaubert et al., 2016). These characteristics make CO relatively chemically inactive. Because of this, there are minimal chemistry effects, where the majority of impacts on CO will result from grid resolution. Fine-scale features of CO are better captured in the ne30x16 horizontal resolution simulations as CO concentrations are more resolved over urban regions, as can be seen in **Fig. 13b**. **Fig. 13c** shows the modeled CO concentrations conservatively regridded from the ne30x16 horizontal resolution to the ne30x8 horizontal resolution. Using this regridding method to go from the finer to the coarser resolution did not reproduce the same results seen from running the ne30x8 simulation (**Fig. 13a**). When the model is run at 1/16˚ horizontal resolution, localized features (e.g., pollution plumes, sharp emission gradients) are better resolved and land use is better represented. . **Figures 13a-13c** have mean CO concentrations over SEMI of 141.7, 131.1, and 132.1 ppb, respectively.

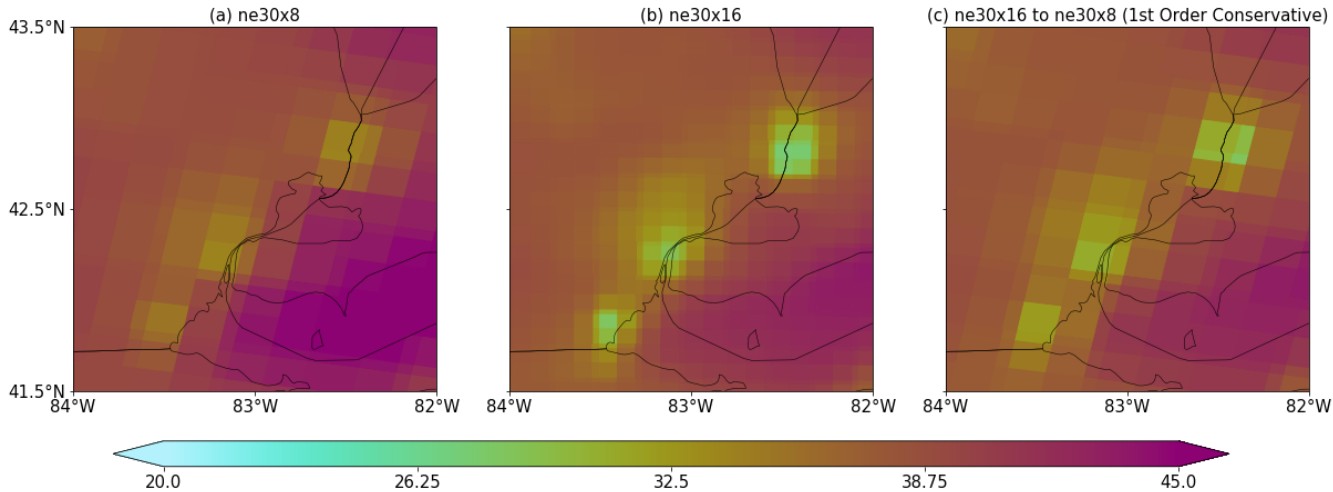


Figure 14: Same as Fig. 13, but for modeled O$_3$ concentrations at the surface.

On the other hand, O$_3$ is highly chemically active. **Figure 14** shows modeled monthly averaged O$_3$ concentrations at the
surface for June 2021 for the ne30x8 (Fig. 14a) and ne30x16 (Fig. 14b), as well as the conservatively regridded model output
from the ne30x16 simulations to the ne30x8 horizontal resolutions (Fig. 14c), respectively. Similarly to what Jo et al. (2023)
found over South Korea, there is a decrease in O$_3$ concentrations over urban areas in SEMI, as a result of NO$_x$ titration. This
reduction in O$_3$ is more prominent with the finer horizontal model grid resolution, which leads to differences in the monthly
mean surface O$_3$ concentrations in coarse (40.1 ppb) and fine (38.0 ppb) horizontal resolutions over SEMI. When regridding
O$_3$ concentrations from the finer to the coarser horizontal resolution, the NO$_x$ titration is visible over the urban areas in SEMI
similarly to the ne30x16 simulation, but is stronger when compared to the ne30x8 simulation.
**Figure 15** shows the diurnal variation for O$_3$, NO, and NO$_2$ concentrations, NO emission flux, and the planetary boundary
layer height from the four simulations presented in this study for three sites – a suburban downwind site (Allen Park), an urban
site (Trinity St. Marks), and a rural site (New Haven). The Allen Park and Trinity St. Marks sites are located within the same
grid box in the ne30x8 simulations, while in the ne30x16 simulations, they are not. **Figures 15a-15c** show that horizontal
resolution had the most impact on O$_3$ concentrations at all sites during peak times (12-18 EDT), with differences between
simulations of up to ~5 ppb. This difference results in an improvement for the ne30x16 simulations based on the findings in
Fig. 4, where peak O$_3$ performed best in the finer resolution simulations when compared to the surface sites. The addition of a
diurnal cycle for anthropogenic NO did not have a significant impact on O$_3$ concentrations during these peak times, but saw
larger differences during the 5-11 EDT timeframe, likely as a result of lower NO (**Figs. 15d-15f**) and NO$_2$ (**Figs. 15g-15i**)
concentrations and associated NO$_x$ titration in the model simulations. It is important that we acknowledge the differences
caused in NO and NO$_2$ at the Allen Park and Trinity St. Marks sites due to grid resolution. As was mentioned before, Allen
Park and Trinity St. Marks are located within the same grid box in the coarser resolution. When using the ne30x16 horizontal
resolution, higher concentrations for NO and NO$_2$ can be seen at the Trinity St. Marks site than at the Allen Park site, which
coincides with the high urbanization in that area. Although these differences are not greatly significant to $O_3$ concentrations,
it is important that urban and non-urban areas are distinguished as they can have higher emissions fluxes (**Figs. 15j-15l**).
On the other hand, a more rural site, like New Haven, has just as much $O_3$ as an urban site, even though NO emission
fluxes are quite low. This is likely a result of the New Haven site being more representative of background concentrations
driven by transport from upwind areas that include close proximity to a major highway and near Lake St. Clair. Hayden et al.
(2011) found that along the Lake St. Clair shore, pollutants can be confined leading to elevated pollutant concentrations and
an increase in oxidizing capacity.


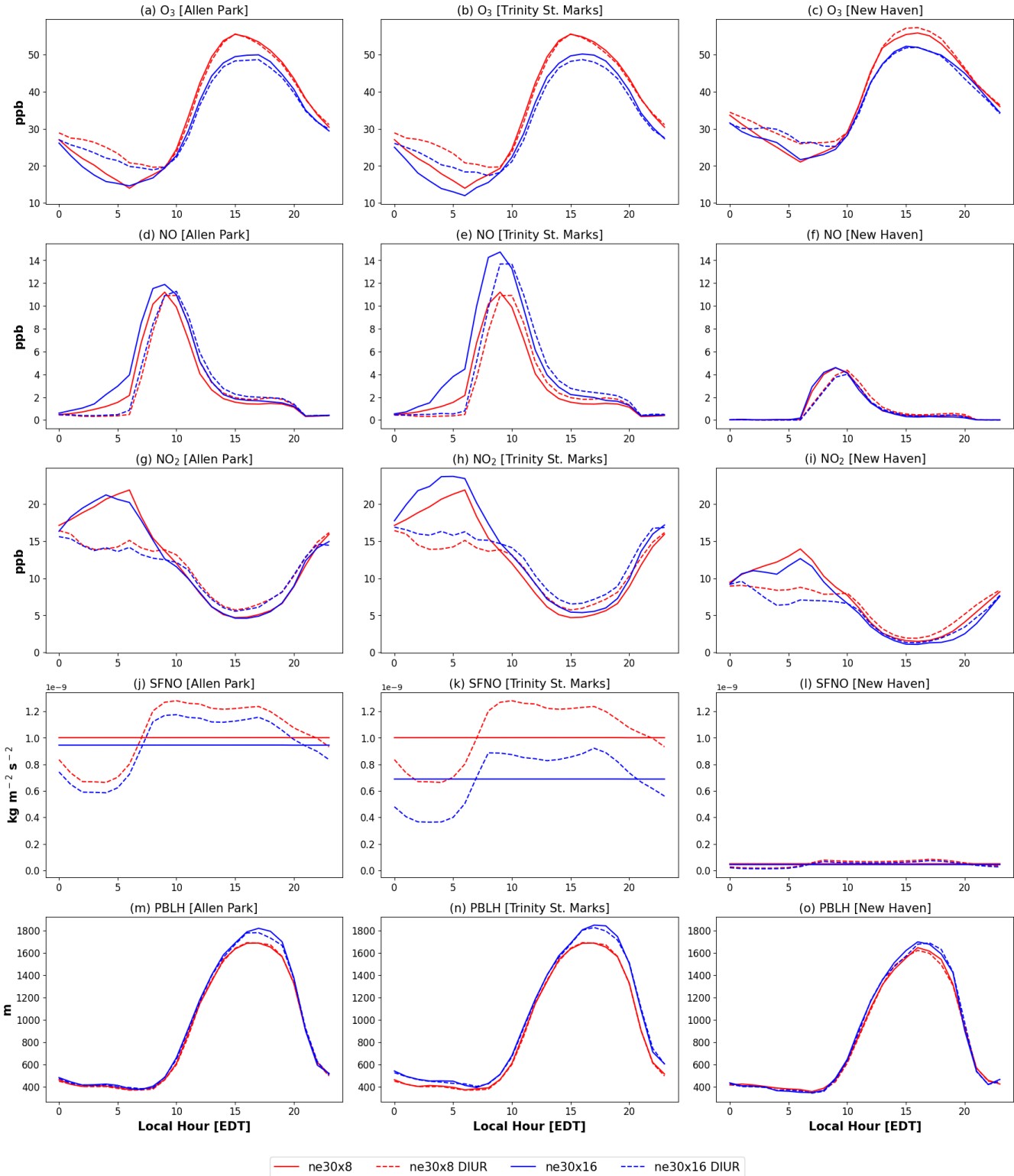


**Figure 15: Diurnal cycle for O$_3$, NO, NO$_2$, NO surface flux (SFNO), and planetary boundary layer heights (PBLH) for three sites in SEMI during the Phase I of the MOOSE campaign [24 May to 30 June 2021] – a suburban downwind site, Allen Park; an urban site, Trinity St. Marks; and a rural site, New Haven. The ne30x8 simulations are represented by the red lines, whereas the ne30x16 simulations are represented by the blue lines. The application of the diurnal cycle to each simulation is represented by the dotted lines for each respective simulation.**

Biogenic VOC emissions can be heavily impacted by changes in model horizontal grid resolution, as they are based on meteorological parameters, such as temperature, as they are calculated online in the land model using the MEGANv2.1 algorithm (see Sect. 2.1.3). **Figure 16** shows isoprene mixing ratios averaged over June and July 2021 from the ne30x8 (**Fig. 16a**) and ne30x16 (**Fig. 16b**) simulations, where the ne30x8 simulation shows about double the amount of isoprene compared to the ne30x16 simulation spread over a wider area. This can be explained by the higher isoprene emission fluxes in SEMI in the coarse resolution (**Fig. 17a**) compared to the finer resolution (**Fig. 17b**). These differences in isoprene emissions caused by the different horizontal resolutions are directly impacting isoprene concentrations in the models. These findings are directly supported by **Fig. 3**, where although temperatures between the simulations are not significantly different, there are changes in cloud totals and winds that could impact solar radiation and thus the isoprene emissions. The differences in temperature between the resolutions are also illustrated in the maps in **Fig. S16-S31** in the SI.

Similarly to O$_3$ and NO$_2$, isoprene has a strong diurnal cycle that is driven by temperature and solar radiation. The diurnal cycles for isoprene, HCHO, and the hydroxyl radical (OH) are shown in **Fig. 18** for the same three sites in **Fig. 15**. For the Allen Park and Trinity St. Marks sites, isoprene mixing ratios (shown in **Figs. 18a-18c**) were generally lower in both simulations, which coincides with suburban and urban landscapes that have relatively low densities of trees, while at the New Haven Site, the concentrations were about double compared to the other two sites as it is a more rural region. For all of the sites, the isoprene concentrations were lower in the ne30x16 simulations compared to the ne30x8 simulations. In the ne30x8 simulations, the isoprene concentrations at the Allen Park and Trinity St. Marks sites are shown to be the same, but when using the ne30x16 resolution, isoprene is shown to be lower in the urban location compared to the suburban location. This indicates that finer resolution can help better characterize regions of interests and assist in the misclassification of emission sources, which coincides with the findings in Sec. 3. The lower isoprene concentrations also impact HCHO concentrations (shown in **Figs. 18d-18f**), as HCHO is a product of isoprene oxidation. HCHO is lower in the ne30x16 simulations, which coincides with the lower isoprene concentrations. OH concentrations (shown in **Figs. 18g-18i**) are generally consistent in all simulations, with very minimal changes after applying the diurnal cycle for anthropogenic NO. Because SEMI is in a more VOC-limited regime (Xiong et al., 2023), OH concentrations are less sensitive to changes in VOCs and more prone to changes resulting from the changing NO$_x$ levels due to titration of O$_3$ (de Gouw et al., 2019). HCHO is NO$_x$ sensitive, meaning that more HCHO is produced in the presence of higher NO$_x$ concentrations (Schwantes et al., 2022). HCHO was not heavily changed by the application of the diurnal cycle for anthropogenic NO emissions, indicating that the main driver for changes in HCHO is grid resolution. In general, isoprene and HCHO decrease with increased resolution, while OH remains relatively constant between model simulations.

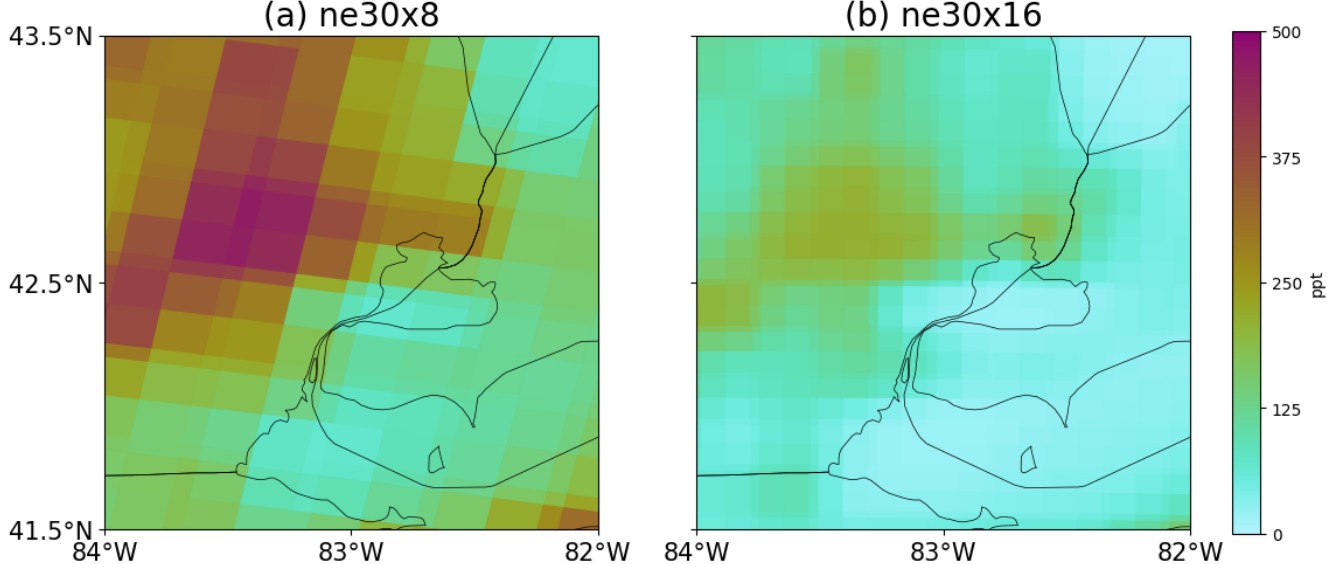


**Figure 16: Modeled isoprene mixing ratios averaged for June and July 2021 over Southeast Michigan.**

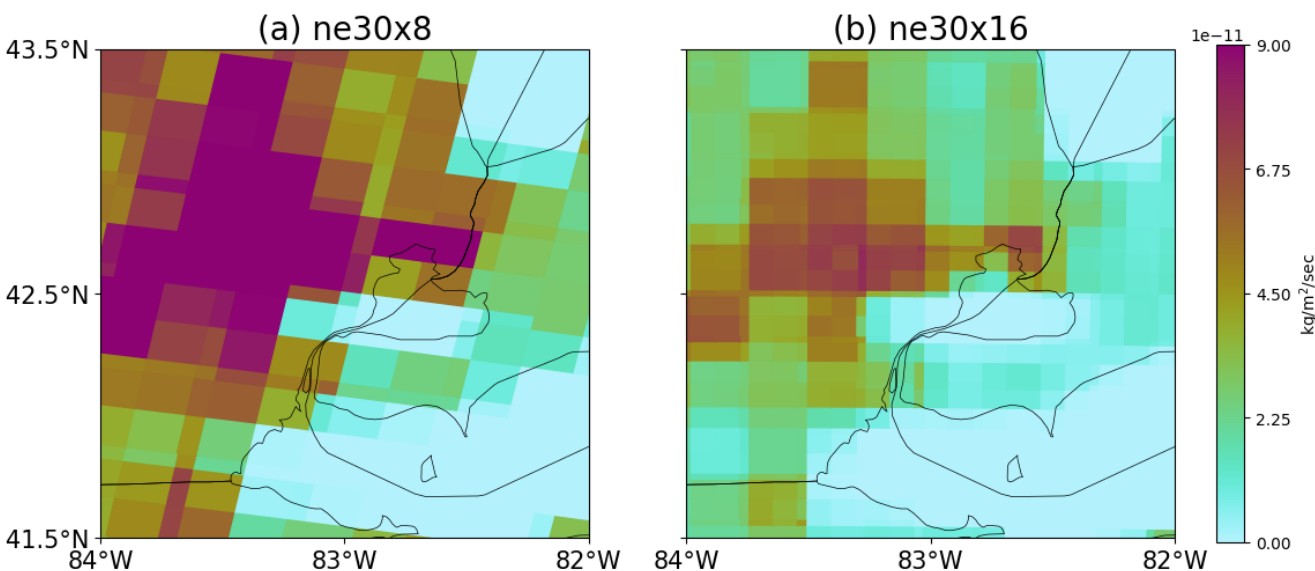


**Figure 17: Isoprene emission flux averaged for June and July 2021 over Southeast Michigan.**

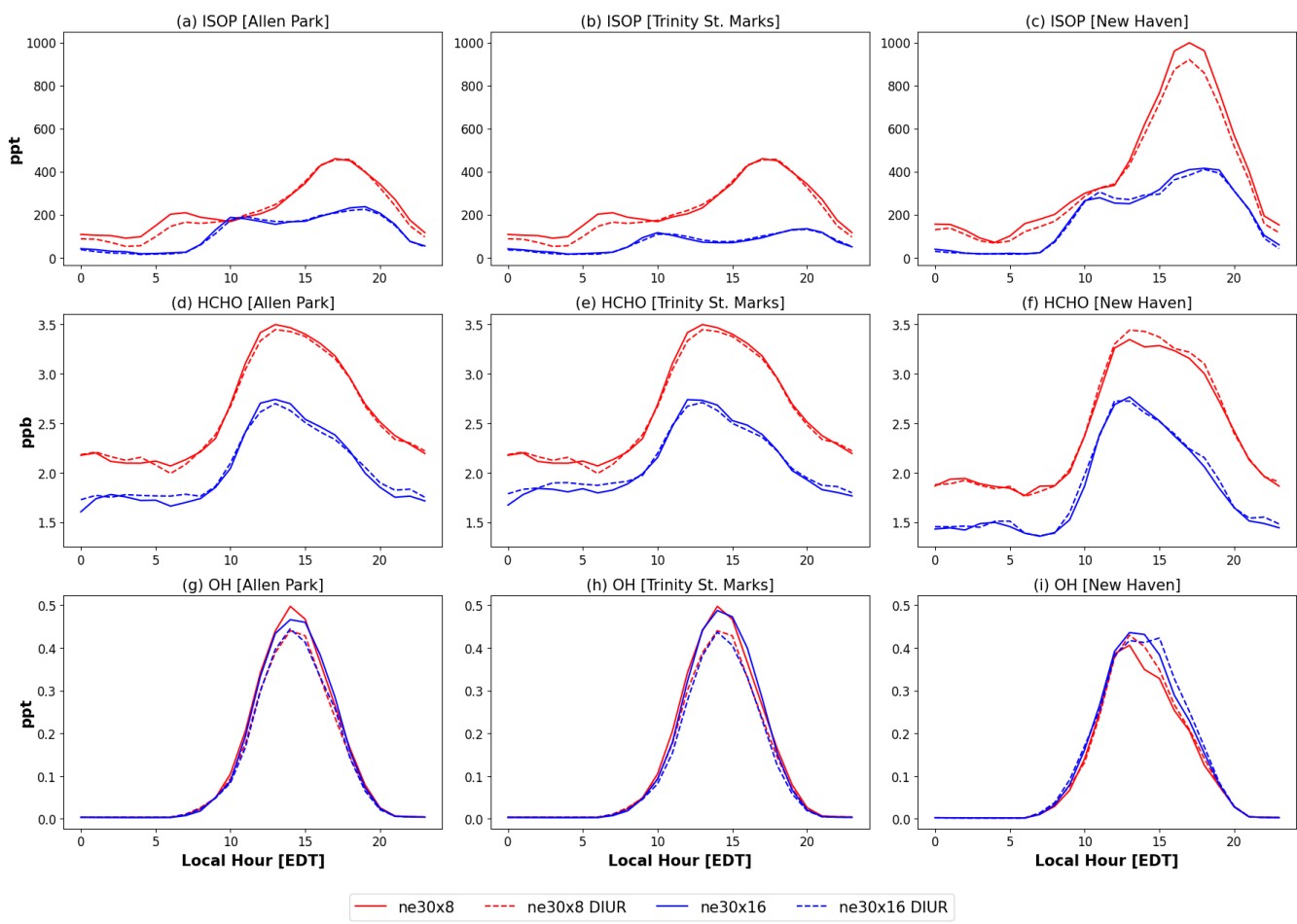

**Figure 18: Same as Figure 15, but for isoprene (ISOP), formaldehyde (HCHO), and the hydroxyl radical (OH).**

The findings of this study show that $O_3$ production in SEMI is strongly governed by the spatial distribution of emissions and different chemical regimes. The urban location analysis showed that Detroit, which is a major industrial hub in the region, is consistent with a VOC-limited regime, where in the daytime, $O_3$ concentrations are suppressed by high $NO_x$ titration, but can become sensitive to changes in VOCs during peak $O_3$ times. The suburban and remote location analysis (i.e., Allen Park and New Haven, respectively) showed that they are in a more $NO_x$-limited regime, where higher BVOCs and lower $NO_x$ titration can lead to more efficient $O_3$ production. The spatial distribution is seen more clearly as we move towards finer resolutions indicating more realistic emissions.

In VOC-limited regimes, targeting reductions in VOCs, such as those from the industrial sectors, is crucial compared to reductions in $NO_x$, as it could lead to temporary increases in $O_3$ production. In $NO_x$-limited regimes, where $NO_2$ drives $O_3$ production, reductions in transportation emissions and long-range transport would decrease $O_3$. The improvement in model representation of $NO_2$ and in turn, $O_3$, during rush hour times (Fig. 4-5) shows how emissions can be misrepresented in the

models. It is necessary that future work considers incorporating higher resolution temporal profile and regional emissions to
better distinguish different $O_3$ processes. Future work should also explore the impacts of targeting the contribution of different
emission scenarios in SEMI to demonstrate the impact of different regulatory decision-making.

**5 Conclusions**

Tropospheric $O_3$ in SEMI is a persistent problem in the region, majorly resulting from anthropogenic activity. MUSICAv0,
a global chemistry-climate model with regional refinement capabilities, has allowed us to evaluate whether the MUSICA
framework is suitable for studying $O_3$ atmospheric chemistry in local, urban environments. This study aimed to evaluate the
impact of horizontal grid resolution and diurnal cycles (for anthropogenic NO emission) on MUSICAv0 simulations, using a
custom grid over the state of Michigan with a resolution of 1/16˚ and leveraging a suite of measurements from Phase I of the
MOOSE field campaign in 2021.
For $O_3$ and its precursors, both grid resolution and the diurnal cycle for anthropogenic NO emissions were important,
largely depending on the time of day and the region. Horizontal grid resolution was important for $O_3$ during peak $O_3$ times
(12-18 EDT), but during the night and early morning, $O_3$ was largely impacted by the application of the diurnal cycle as a
result of changing $NO_2$ during $NO_x$ peak times.
This work compares simulated $NO_2$ and HCHO tropospheric columns from MUSICAv0 model runs to measurements
from the Pandora Network at two sites in SEMI for the first time. $NO_2$ columns from Pandora agreed with the temporal
variability of $NO_2$ columns at the two urban sites, where the application of the diurnal cycle for anthropogenic NO emissions
at both resolutions generally made the model perform better during peak $NO_2$ times, but saw greater model overestimations in
the later part of the day. These trends are important as they can be indicative of high anthropogenic emission sources from
industry and transportation in the model. Modeled HCHO columns compared to Pandora, on the other hand, were largely
impacted by a combination of grid resolution and the diurnal cycle, where grid resolution impacts HCHO because of online
calculations of biogenic VOCs and changing $NO_x$ levels can promote VOC oxidation leading to lower HCHO columns in the
model. These changes led to underestimations of HCHO tropospheric columns in the model compared to observations. This
underestimation indicates that the model simulations are not capturing anthropogenic VOC efficiently.
In addition, $NO_2$ tropospheric columns from the model simulations were compared to observations from GCAS for the
first time. This comparison showed that the finer resolution captured more pronounced pollution plumes corresponding to
observed wind directions, which can be important when assessing transport from more localized sources. As grid resolution in
global chemistry-climate models is becoming finer, future work should consider using $NO_2$ column data from remote sensing
instruments to develop regional emission inventories for more fine-scale applications.
This work showed that grid resolution is more important for $O_3$ precursors (i.e., $NO_x$, HCHO, isoprene) than for $O_3$ itself,
which agrees with the findings in Jo et al. (2023) and Schwantes et al. (2022). Changes due to grid resolution were largely a
result of the artificial mixing of emissions. Finer resolutions can better classify source regions and distinguish between urban

and non-urban regions. Grid resolution also impacted biogenic VOCs, as they are calculated online via MEGANv2.1 based on various meteorological parameters. Although isoprene in the finer resolution simulations showed better performance compared to the AML measurements, SEMI is generally not prone to high isoprene emissions. Future work using the regional refinement grid over Michigan should focus on evaluating locations with higher vegetation density.

Applying diurnal cycles for anthropogenic NO on monthly emissions also played a crucial role in nighttime $O_3$ chemistry. The diurnal cycle often impacted $O_3$ and precursor concentrations more than grid resolution. Future work should evaluate the impacts of applying diurnal cycles to more anthropogenic emissions, other than NO. In addition, we acknowledge that apart from applying a diurnal cycle for anthropogenic NO emissions, the evolution of the PBL can also play a significant role in the formation of $O_3$ and $NO_x$. In the daytime, a rising PBL can mix surface $NO_x$ and VOCs upwards, reducing $O_3$ concentrations near the surface, while in the nighttime, a shallower PBL can trap emissions near the surface leading to higher $NO_x$ titration. Uncertainties associated with the PBL could lead to underpredictions of $NO_2$ in the model and misrepresentations of $O_3$ peaks.

This is one of a few studies evaluating $O_3$ production and loss processes with custom grids in MUSICAv0. Although $O_3$ biases still persist in the MUSICAv0 simulations over this region, these biases are generally lessened with finer grid resolution during peak $O_3$ times, and with diurnal cycles for anthropogenic NO during the nighttime. This case study is limited to SEMI, which can have different implications compared to previous work. For example, the state of Michigan is about 2.5 times larger than South Korea, which was studied in Jo et al. (2023) using a similar methodology, and has a completely different topography. Michigan is generally flat and surrounded by freshwater lakes, as opposed to Korea's mountainous terrain surrounded by ocean encompassing a megacity. Schwantes et al. (2022) found that $O_3$ was better simulated over urban regions across the Southeastern US, especially when using a ~14 km regional refinement grid and updated chemistry in MUSICAv0. This work took into consideration a finer grid resolution mesh (~7 km) and compared to ~14 km to show that regional refinement improves $O_3$ representativeness in the model. Future work aims to take advantage of custom grids to quantify the contribution of emissions and transport on $O_3$ atmospheric chemistry in the region. Future work should also take into consideration the use of a more updated version of the CAMS-GLOB-ANT emissions, as well as the diurnal variation profiles of CAMS-GLOB-TEMPO (Guevara et al., 2021; Soulie et al., 2024), or more regional emission inventories such as the National Emission Inventory (NEI) from the US EPA. Optimization of a regionally-refined, coupled model such as MUSICAv0, through resolution and emission modeling studies, can have significant implications for the design and development of effective surface $O_3$ mitigation strategies in SEMI.

**Code and Data Availability**

Aircraft and mobile laboratory measurements during the MOOSE field campaign are freely available at the NASA data archive: https://www-air.larc.nasa.gov/missions/moose/. Surface measurements for the state of Michigan can be found at Michigan's Department of Environment, Great Lakes, and Energy Air Monitoring Site: https://www.michigan.gov/egle/about/organization/air-quality/air-monitoring. Data from the Pandonia Global Network can be

found at: https://data.ovh.pandonia-global-network.org/. CESM2.2 (which includes MUSICAv0) is an open-source community model available at: https://github.com/ESCOMP/CESM, with the code version including application of diurnal variation for emissions at: https://doi.org/10.5281/zenodo.8044736 (Jo, 2023). The CAMS-GLOB-ANTv5.1 and CAMS-GLOB-AIRv2.1 emission inventories is available at the ECCAD database (https://eccad.sedoo.fr/). The grid information files for the custom grid mesh over Michigan and processed model output are available at https://doi.org/10.5281/zenodo.14625128 (Mariscal, 2025).

## Supplemental Information

The supplement related to this article is available online at: https://acrobat.adobe.com/id/urn:aaid:sc:us:58d479d2-b091-4aa1-a1be-0129b5661241

## Author Contributions

NM, LKE, and YH were involved in the overall design and execution of the study. NM constructed the grid mesh over Michigan, prepared input datasets, ran the model simulations, and led the analysis. DSJ provided the source code modifications for adding diurnal cycle of emissions, code for regridding input datasets and model output, and code for processing model results. JC conducted measurements during MOOSE. YX, LMJ, SJJ, and other coauthors provided thorough discussions on the study. NM prepared the manuscript with improvements from all coauthors.

## Competing Interests

The authors declare that they have no conflicts of interest.

## Disclaimer

## Acknowledgements

The authors would like to acknowledge the entire MOOSE field campaign teams, especially Lukas Valin at the US Environmental Protection Agency for guidance on processing Pandora spectrometer data, and Aerodyne Research, Inc. scientists and technicians who participated in the MOOSE mobile field measurements and data quality assurance: Tara Yacovitch, Brian Lerner, Francesca Majluf, and Manjula Canagaratna. In addition, we would like to acknowledge the US EPA Office of Research and Development for the Pandora data used from the SWDetroitMI and DearbornMI locations, as well as the NASA Goddard Space Flight Center and Luftblick for Pandora calibration and data products. We thank Pablo Lichtig for sharing their postprocessing scripts, which served as a foundation for the tropospheric column analysis. We would like to

acknowledge the high-performance computing support and data storage from the Cheyenne (DOI:10.5065/D6RX99HX) and
Derecho (DOI:10.5065/qx9a-pg09) supercomputers provided by NSF NCAR and sponsored by NSF.

## Financial Support

This work was supported by the National Science Foundation (NSF), with the Grant Nos. of 2111428, 2126097 and 1735038.
This material is based upon work supported by the NSF National Center for Atmospheric Research (NCAR), which is a major
facility sponsored by the NSF under Cooperative Agreement No. 1755088. Duseong S. Jo was supported by the New Faculty
Startup Fund from Seoul National University.

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
