# Peer review of "Evaluation of Ozone and its Precursors using the Multi-Scale"

_EGUsphere, 2025_

## Author Comment (AC1)

**Response to Reviewer #1:**

We would like to thank the reviewer for their careful and thorough reading of this manuscript and for the thoughtful and constructive comments and suggestions, which will help improve the overall quality of this manuscript. Our responses are denoted in **red**.

**General Comments:**

The manuscript analyzes the distribution of ozone and some of its precursors over the Southeast Michigan (SEMI) region during summer 2021 based on model simulations with MUSICAv0 and observations from the MOOSE filed campaign. The authors discuss the impact of grid resolution and diurnal cycle of anthropogenic NO emissions and show that night-time ozone is mostly improved by applying diurnal cycles for NO emissions, while grid resolution is found to have more impact on ozone precursors. The study also shows that using a good conceptualization of grid resolution within MUSICAv0, with finer resolution could lead to more efficient computational costs, which could be beneficial for other local-scale studies including in other regions.

The paper shows the interesting potential of using global models with zooming capabilities like MUSICAv0 to investigate air pollution characteristics even at specific small regions like SEMI. Overall, the paper is well structured and easy to read. However, the analysis and discussion sections are in some cases rather short and could be further improved in order to better identify the processes controlling summertime ozone in different parts of the SEMI region.

I recommend the manuscript to be accepted for publication after addressing the following comments and suggestions:

Response: We thank Reviewer #1 for their careful observations, and appreciate their feedback and recommendations for improving the manuscript. We have carefully gone through all of your comments, and have addressed them below and in the main text.

**Section 2.1.1:** Initial conditions are considered from a restart file based on MOZART-TS1. Which initial conditions are considered for the additional species in TS2 not included in TS1?

Response: We thank the reviewer for their observation. We have added a sentence addressing the usage of the initial condition file with TS1 for TS2 simulations in Section 2.1.1: "Although the initial condition file was based on MOZART-TS1 chemistry and the additional species in MOZART-TS2 were initiated from zero, the majority of these species are short-lived and equilibrate quickly within the one-month spin-up period."

**Section 2.1.3:**

- Anthropogenic emissions are considered from CAMS_GLOB_ANTv5.1. A recent study from Soulie et al. (2024, ESSD) shows significant differences in the estimated emissions between CAMS_GLOB_ANTv5.1 and the EPA inventory in USA. In particular, EPA exhibits higher NMVOCs but lower NOx and SO2 emissions. Can the authors comment on the potential impact of such uncertainties in emissions on the model results?

Response: Yes, we acknowledge the discrepancies between CAMS_GLOB_ANTv5.1 and the US EPA's National Emission Inventory (NEI) described in Soulie et al. (2024). The differences in the emissions, especially the high NMVOCs and low NOx and SO2, in NEI compared to CAMS definitely has the potential to introduce uncertainties in the model results because emissions directly influence atmospheric chemistry and pollutant concentrations. Increased availability of NMVOCs could lead to increased O3 production, especially in VOC-limited regimes (i.e., more urban areas; O3 decreases with increase in NOx and increases with increase in VOC) and potentially alter the oxidizing capacity of the atmosphere. On the other hand, in NOx-limited regimes (i.e., more rural areas; O3 increases with increases in NOx and changes very little with changes in VOC), the lower availability of NOx could reduce O3 concentrations in the model. While these uncertainties reflect broader challenges pertaining to emission inventories such as spatial distribution and sectoral estimates, our study uses CAMS for consistency with other global studies, and explores the impact of relative changes such as adding diurnal variation of emissions. Future work should include the use of regional inventories, such as NEI, or inventories derived from inverse modeling. In addition, the lower emissions for NOx and SO2 could also alter secondary aerosol formation (e.g., lower NOx could decrease nitrate aerosol formation). Although this is a non-negligible issue, it is out of the scope of this study and should be addressed in future work.

We included a sentence commenting on future considerations in Section 5 as so: "Future work should also take into consideration the use of a more updated version of the CAMS-GLOB-ANT emissions, as well as the diurnal variation profiles of CAMS-GLOB-TEMPO (Guevara et al., 2021; Soulie et al., 2024), or more regional emission inventories such as the National Emission Inventory (NEI) from the US EPA."

- It is not clear how soil NOx emissions are considered for the simulations.

Response: Global soil NOx emissions are based on the natural emissions of NO as described in Emmons et al. (2020). This is now mentioned in Section 2.1.3 as so: "Other emissions, from soil, lightning, volcanoes and oceans, are described in Emmons et al. (2020)."

- The authors include calculated NO emissions from agriculture waste burning (AWB) in Table S1, but it is not clear if emissions from this sector are considered or not. This could lead to double counting of emissions with QFED, although the contribution of NO AWB emissions seems to be minor compared to other sectors.

Response: CAMS anthropogenic emissions from all available NO sectors are considered when applying the diurnal cycle, where we've used a sector-based and country-specific temporal profile. There is a possibility that agricultural waste burning (AWB) emissions could be double counted in the emissions, as QFED uses satellite observations of the fire radiative power (i.e., rate of radiative energy emitted by an active fire) to estimate global gridded fire emissions, but for our region of study, this impact is minimal. Regardless, we have added a footnote in Table S1 of the Supplemental Information to note this uncertainty (see below).

**Table S1: Total anthropogenic nitric oxide (NO) emissions from the CAMSv5.1 in Michigan and Southeast Michigan, and their ratio**
**from various sectors\*.**

|  | MICH [kt] | SEMI [kt] | SEMI/MICH |
|---|---|---|---|
| AGS | 0.46 | 0.04 | 9.0% |
| AWB[1] | 0.10 | 0.01 | 10.8% |
| ENE | 9.44 | 2.86 | 30.3% |
| RES | 1.03 | 0.48 | 47.1% |
| TNR | 1.99 | 0.36 | 18.1% |
| TRO | 15.13 | 3.50 | 23.2% |

\*AGS = Agriculture Soils; AWB = Agriculture Waste Burning;
ENE = Power Generation; RES = Residential; TNR = Off-Road
Transportation; TRO = Road Transportation
[1]It is possible that AWB emissions could be double counted via
biomass burning emissions from QFED (QuickFire Emissions Dataset),
where the fire radiative power obtained from the satellite is used to
estimate the global gridded fire emissions (Darmenov and da Silva,
2015). Although it has been found that AWB can increase fire
emissions over regions, in Southeast Michigan this contribution is
minimal.

**Section 2.1.4:**

-  Can the authors comment why only NO diurnal distribution is considered, while diurnal distribution
of other species like VOCs or SO2 could also impact the model results?

Response: We acknowledge that the diurnal distribution of VOCs and SO2 could impact model results,
especially for O3 chemistry. In our study, we focus on applying the diurnal cycle for anthropogenic NO
emissions due to its dominant role in controlling tropospheric O3 and titration processes, which are highly
sensitive in areas dominated by industrial and transportation-based activities, like SEMI. We recognize
that applying diurnal variation for VOCs and SO2 could also affect O3 production as they can be
temperature-driven (e.g., biogenic VOCs) and based on industrial activity. Future work will incorporate
temporal profiles for all available anthropogenic emissions, as they would help refine model results and
further assess the impact on other critical air pollutants.

We have added text to Section 2.1.4: "While emissions of other anthropogenic compounds, such as VOCs,
do have diurnal variations, we have only implemented the diurnal variation for NO emissions in this work,
due to its dominant role in controlling tropospheric O3 and titration processes."

-  Including a figure showing the diurnal distribution of NO emissions from different sectors, as used in
the simulation is a useful information.
Response: We have included a figure of the diurnal profiles for each sector in Fig. S2 in the supplemental
information of the manuscript and reference it in Section 2.1.4.

[Figure]

**Figure S2: The diurnal variation scale factors applied to NO emissions for each anthropogenic emission sector used**
**in the simulations.**

**Section 3.1:**

-  This section is rather short and doesn't fully cover the model's ability to capture meteorological
features in the considered region. In addition to the model evaluation, this section is also expected to
contain a description of the meteorological situation that characterized the SEMI/MI region during
the campaign period. This section can also be significantly improved by considering other
meteorological variables (e.g. wind speed/direction), other networks or datasets (e.g. reanalysis).

Response: We have expanded this section in the main text to elaborate further on the campaign period and the presented figure. We have also included a time series comparison of the AML wind speeds and wind directions to better support this section.

Expanded Text: "SEMI is a region that faces unique air quality challenges due to large industrial and automotive activity, dense population, and geographic factors. SEMI has a diverse terrain, ranging from highly urbanized areas, such as the city of Detroit, expansive agricultural lands in more remote areas, and forests surrounded by both inland and coastal lakes. The region consists of a relatively flat terrain, with a humid continental climate. Additionally, large air masses of humidity can be transported into the region from the Great Lakes (i.e., Lake Huron and Lake Erie) through the lake effect winds (Scott and Huff, 1996). A time series along the AML track of meteorological values – temperature, relative humidity, planetary boundary layer height, cloud total, wind speed, and wind direction – from the models (and observations for temperature and relative humidity) are shown in **Fig. 3**. During the campaign period in the summer of 2021, temperatures reached up to approximately 305 K and relative humidity to almost 100%. The planetary boundary layer reached more than 2500 m on most days, while cloud total was relatively varied. Modeled wind speeds follow the trend for the campaign period quite well, but are comparatively high compared to the observations, while wind directions perform generally well except on some specific days. The AML track covered a large part of the SEMI region, making its way through both very urban and rural areas. Meteorological parameters, such as temperature, are highly impacted by urbanization through the reductions in vegetated land cover and increases in energy consumption (Wang et al., 2021). Urbanization can lead to higher temperatures, and thus increasing $O_3$ production. In the simulations presented here, meteorological parameters (i.e., temperature and horizontal winds) are nudged towards reanalysis data to obtain a more realistic depiction of reality in the coarser resolution regions, leaving the regional refinement area to freely run, as the resolution of the refined area is finer than the resolution of the reanalysis dataset that is being used. Regional refinement grids, with high horizontal resolution, are capable of resolving areas with large geographical differences (Jo et al., 2023). Meteorological fields in these simulations are generally consistent indicating that meteorology is performing similarly, even with the changes in horizontal resolution. Although temperatures, relative humidity, and planetary boundary layer height remain consistent among all the simulations, cloud total varies between the simulations, which can significantly impact photochemical production."

Updated Figure:

[Figure]

**Figure 3: Time series of (a) temperature, (b) relative humidity, (c) planetary boundary layer height, (d) cloud total, (e) wind speed,**
**and (f) wind direction along the Aerodyne Mobile Laboratory (AML) track. Measurements of temperature and relative humidity**
**were available and displayed as black x's in Fig. 3a and 3b. The model results are shown in red (ne30x8) and blue (ne30x16)**
**corresponding to horizontal resolutions. The dashed lines represent model simulation results when adding the diurnal cycle for nitric**
**oxide anthropogenic emissions, color-coded to their respective horizontal resolution.**

**Section 3.2:**

- The authors could elaborate a bit more the discussion on the reasons behind the diurnal changes in the model bias and link with results in Sect. 4. For this, a map showing location of the stations could be very useful.

Response: A map that includes the stationary site locations for this evaluation can be found in Fig. 2 of the manuscript. A stronger link has been made between Sec. 3 and 4 to better enhance the discussion.

[Figure]

**Figure 2: Location of observations from Phase I (24 May to 30 June 2021) of the Michigan-Ontario Ozone Source Experiment (MOOSE) used in this study. The gray line shows the track of the Aerodyne Mobile Laboratory across Southeast Michigan. Stationary sites from the Michigan Department of Environment, Great Lakes, and Energy (MI EGLE) are shown as the red numbers (1-12), and the Pandora monitoring sites are shown as the yellow letters (A-B).**

These linkages are reflected in Section 4 as so:

- "This difference results in an improvement for the ne30x16 simulations based on the findings from in Fig. 4, where peak $O_3$ performed best in the finer resolution simulations when compared to the surface sites."
- "These findings are directly supported by **Fig. 3**, where although temperatures between the simulations are not significantly different, there are changes in cloud totals and winds that could impact solar radiation and thus the isoprene emissions. The differences in temperature between the resolutions are also illustrated in the maps in **Fig. S16-S31 in the SI**."

- The night-time NO2, in particular between 00 and 05 AM, although improved, remain high and the morning peak is less visible when NO diurnal cycle is applied. The authors should discuss the impact of potential uncertainties in the considered diurnal cycle, including the fact that this was applied only for NO.

Response: We thank the reviewer for their insightful observation regarding the nighttime NO2 concentrations and the less prominent morning peak when the diurnal cycle for anthropogenic NO emissions is applied in the model. We acknowledge the potential uncertainties in the presented diurnal cycle and have expanded our discussion to take this into account.

- Possible causes for the high nighttime NO2 concentrations include an overestimation of nighttime NO emissions. While the temporal profile applied to NO scales the anthropogenic emissions, some emission sectors (i.e., ENE, AWB; see temporal profile figure) have flatter temporal cycles that could lead to the sustained concentrations of NO2 via the reaction of NO and O3. Nighttime NO2 can also accumulate due to reduced O3 titration, as well as due to a shallow nighttime boundary layer that can trap NOx concentrations.
- The less prominent peak could be due to the temporal profile application to anthropogenic NO emissions, as the morning rush may be too gradual, so the NO2 peak appears more delayed than the observed values. In addition, the morning VOC emissions from transportation-related activities could further enhance the NO-to-NO2 conversion, but since VOC diurnal cycles are not included, this feature could be underrepresented.

We have added a few sentences stating this uncertainty in Section 5 of the main text: "In addition, we acknowledge that apart from applying a diurnal cycle for anthropogenic NO emissions, the evolution of the PBL can also play a significant role in the formation of $O_3$ and $NO_x$. In the daytime, a rising PBL can mix surface $NO_x$ and VOCs upwards, reducing $O_3$ concentrations near the surface, while in the nighttime, a shallower PBL can trap emissions near the surface leading to higher $NO_x$ titration. Uncertainties associated with the PBL could lead to underpredictions of $NO_2$ in the model and misrepresentations of $O_3$ peaks."

**Section 3.3:**

- The authors relate the differences in simulated isoprene (and hence HCHO?) to potential changes in meteorological field leading to changes in calculated BVOCs. Although this could be true, no results (i.e. changes in meteorology) are provided to assess this especially in the discussion in Sect. 3.1.

Response: Thank you for pointing out this discrepancy between the sections 3.1 and 3.3. As mentioned in an earlier part of this review, some necessary corrections and additions to section 3.1 have been made. Based on this, we have added some further explanation on what else could be impacting simulated isoprene. We have added discussion of how the biogenic emissions could be impacted by changes in cloud totals between the simulations.

This new addition to section 3.3 is: "Although temperatures are not greatly affected by grid resolution, as was seen in **Fig. 3**, cloud totals are different in the two resolutions, which can impact the amount of solar radiation reaching the surface. Clouds in the model can be impacted by several changes, such as changes in aerosols, which is out of the scope of this study, or related to changes in meteorology (e.g., winds). Yan et al. (2023) demonstrated that aerosols are able to impact precursor accumulation and photolysis (e.g., isoprene), where tropospheric chemical loss is enhanced due to photolysis and $NO_x$ accumulation. Cheng et al. (2022) also found that changing clouds in chemical transport models can impact photochemical reaction rates and BVOCs. Future work on evaluating model grid resolution and diurnal cycle impacts on $O_3$ formation should look more closely into aerosol-cloud interactions and how they impact photochemical production in SEMI."

- Similarly, the discrepancies in other species (hydrocarbons and aromatics) is explained by
misrepresentation of their anthropogenic sources in the CAMS inventory. The authors can assess such
uncertainties in the considered emissions by comparisons with EPA emissions in SEMI.

Response: We have made a quick comparison of CAMSv5.1 emissions with HTAP_v3.1 mosaic, which
includes emissions from the US National Emission Inventory (NEI) (Crippa et al., 2023). Because
HTAP_v3.1 is only available up until 2020, we have compared with CAMSv5.1 for 2020 to illustrate the
misrepresentation of emissions based on the emission inventory without year-to-year discrepancies. We
have also included emissions for 2021 from CAMSv5.1 to show the differences between years. The
emission totals shown in this table are representative of summertime emissions (June, July, August) for a
domain over the state of Michigan (longitude: 273˚W to 278˚W; latitude: 41.5˚N to 46˚N). The
comparison of CAMSv5.1 to HTAP_v3.1 shows large differences, especially when comparing the energy,
fugitives, solvents, road transport, and residential sectors. NEI is more regionally representative of the
United States, so taking these differences into account would provide different results in the model
simulations.

| Naming Convention | | Emission (in kt) | | |
|---|---|---|---|---|
| CAMS | HTAP | CAMSv5.1 (2021) | CAMSv5.1 (2020) | HTAP_v3.1 (2020) |
| ene | Energy | 5.223 | 5.254 | 0.327 |
| ind | Industry | 7.683 | 7.715 | 4.484 |
| fef | Fugitives | 9.557 | 9.651 | 7.241 |
| slv | Solvents | 15.548 | 15.529 | 29.011 |
| tro | Road Transport | 22.294 | 23.23 | 9.197 |
| tnr | Other Ground Transport | 0.365 | 0.39 | 1.042 |
| res | Residential | 1.856 | 1.887 | 12.002 |
| swd | Waste | 0.004 | 0.004 | 1.672 |
| awb | Agricultural Waste Burning | 1.007 | 1.005 | 0.102 |
| agl | Agriculture Livestock | 1.898 | 1.889 | 2.062 |
| - | Agriculture Crops | - | - | 3.274 |
| shp | International Shipping | 0.025 | 0.025 | 0 |
| | Domestic Shipping | | | 0.023 |
| | SUM | 65.46 | 66.579 | 70.437 |

Crippa, M., Guizzardi, D., Butler, T., Keating, T., Wu, R., Kaminski, J., Kuenen, J., Kurokawa, J., Chatani, S., Morikawa, T., Pouliot, G., Racine, J., Moran, M. D., Klimont, Z., Manseau, P. M., Mashayekhi, R., Henderson, B. H., Smith, S. J., Suchyta, H., Muntean, M., Solazzo, E., Banja, M., Schaaf, E., Pagani, F., Woo, J.-H., Kim, J., Monforti-Ferrario, F., Pisoni, E., Zhang, J., Niemi, D., Sassi, M., Ansari, T., and Foley, K.: The HTAP_v3.1 emission mosaic: merging regional and global monthly emissions (2000–2018) to support air quality modelling and policies, Earth Syst. Sci. Data, 15, 2667–2694, doi:10.5194/essd-15-2667-2023, 2023.

**Section 3.4:**

- The discussion on evaluation of modeled HCHO columns contradicts the conclusion in Section 3.3: the authors say there is a combined effect of grid resolution and application NO diurnal cycle on HCHO (Line 429), whereas Sect. 3.3 states no obvious impact of NO on HCHO in the model (Line 383).

Response: This has been corrected in the main text as "The differences in grid resolution are seen more strongly than the inclusion of diurnal NO emissions for HCHO concentrations in **Fig. 6b**" in Section 3.3 and the removal of "Because HCHO does not have an obvious diurnal cycle and is different from $NO_2$, performance for HCHO columns was much more dependent on the combined effect of grid resolution and the application of the diurnal cycle for anthropogenic NO" in Section 3.4.

- The section could be improved by discussing the link between the location of the stations/sites and the changes in HCHO (e.g. induced impact from isoprene emissions under different Nox-regimes).

Response: The locations are relatively close to each other, in urban/near-urban areas surrounded by industry. Additional text has been added in Section 3.4 as so:

- "We compare $NO_2$ and HCHO tropospheric columns from two Pandora spectrometers to the four MUSICAv0 simulations. Both Pandora monitoring sites (SWDetroitMI and DearbornMI) were located in an industrial and high-traffic setting, providing continuous observations in urban conditions and complementing the other observations"

- "The locations of the Pandora spectrometers are in highly industrialized, urban areas. The large model bias in HCHO columns could be an indication of missing emission sources in the area."

- Like for other Sect. 3 subsections, it would be useful to include the location of the monitoring sites and link the results with those discussed in Sect. 4.

Response: A map that includes the Pandora sites shown for this evaluation can be found in Fig. 2 of the manuscript (can also be seen in the response for Section 3.2 above). A stronger link between Sec. 3 and 4 has been included in the main text. We've added a transition paragraph at the end of Section 3, which is shown below.

"Section 3 has evaluated the model simulations against four different types of observations obtained during MOOSE 2021. Taken together, the model evaluation shows (i) that refining the horizontal grid resolution in the model is the dominant factor leading to reductions in bias for peak $O_3$ concentrations, enhances $NO_2$ source region plumes, and better separates contrast between urban and suburban locations, such as Allen Park and Trinity St. Marks; (ii) that the diurnal cycle for anthropogenic NO emissions corrects the early morning biases in $NO_2$ and slightly impacts $O_3$, while having small impacts on peak $O_3$ values; and (iii) the high biases in VOCs points to deficiencies in the emission inventory rather than grid resolution and temporal allocation. These findings motivate the more in-depth analysis described in Sec. 4, where we discuss resolution- and diurnal emission-driven changes governing $O_3$ production and loss across SEMI."

**Section 3.5:**

- Surface maps for winds, temperature and other meteorological parameters could be added to the Supplement to better understand the conditions during the analyzed days and times.

We included a table with detailed information of each GCAS raster on the flight days during MOOSE 2021 in the supplemental information as Table S8. We have also included maps during the GCAS flights of temperature and winds in the supplemental information as Figures S16-S31.

**Section 4:**

- The significant changes in isoprene emissions from MEGANv2.1 depending on the grid resolution is linked to the induced changes in meteorological parameters. This needs to be supported by maps of meteorological fields showing these changes.

Response: We thank the reviewer for this suggestion. We have incorporated a couple of sentences referencing Figures 3 and S16-S31, where different meteorological parameters are shown. We have noted some of the differences in relation to how they can impact isoprene. This is noted in Section 4 as: "These findings are directly supported by **Fig. 3**, where although temperatures between the simulations are not significantly different, there are changes in cloud totals and winds that could impact solar radiation and thus the isoprene emissions. The differences in temperature between the resolutions are also illustrated in the maps in **Fig. S16-S31** in the SI."

- The link between Sect 4. and Sect 3. should be strengthened in either or both sections to better understand what drives the changes in the different sites, locations, etc.

Response: We have strengthened the link between sections 3 and 4 to better understand the changes driving O3 chemistry in the region. We have added a transition paragraph at the end of Sec. 3, and an updated introduction for Sec. 4. We have also linked some of the findings in Sec 4. with what was found in Sec. 3. Line 612 is an example of this: "This difference results in an improvement for the ne30x16 simulations based on the findings in Fig. 4, where peak $O_3$ performed best in the finer resolution simulations when compared to the surface sites."

-   The discussion section is rather short and could/should be improved by strong arguments on e.g. what controls O3 in different parts of the SEMI region and what mitigation strategies could be adopted to reduce the pollution.

Response: We have elaborated further on what is controlling O3 in diverse locations across SEMI and incorporated some potential mitigation strategies, as well as future work using this modeling framework at the end of Sec. 4. The added contents in the updated manuscript are shown as follows.

"The findings of this study show that $O_3$ production in SEMI is strongly governed by the spatial distribution of emissions and different chemical regimes. The urban location analysis showed that Detroit, which is a major industrial hub in the region, was consistent with a VOC-limited regime, where in the daytime, $O_3$ concentrations are suppressed by high $NO_x$ titration, but can become sensitive to changes in

VOCs during peak $O_3$ times. The suburban and remote location analysis (i.e., Allen Park and New Haven, respectively) showed that they were in a more $NO_x$-limited regime, where higher BVOCs and lower $NO_x$

titration can lead to more efficient $O_3$ production. The spatial distribution is seen more clearly as we move towards finer resolutions indicating more realistic emissions.

In VOC-limited regimes, it is necessary to reduce emissions of VOCs as well as NOx to avoid increasing

$O_3$ concentrations. Thus, targeting reductions in VOCs, such as those from the industrial sectors, is crucial. In $NO_x$-limited regimes, where $NO_2$ drives $O_3$ production, reductions in transportation emissions and long-range transport would decrease $O_3$. The improvement in model representation of $NO_2$ and in turn, $O_3$, during rush hour times (**Fig. 4-5**) shows how emissions can be misrepresented in the models. It is necessary that future work considers incorporating higher resolution temporal profiles and regional emissions to better distinguish different $O_3$ processes. Future work should also explore the impacts of targeting the contribution of different emission scenarios in SEMI to demonstrate the impact of different regulatory decision-making."

---

## Author Comment (AC2)

**Response to Reviewer #2:**

We would like to thank the reviewer for their careful and thorough reading of this manuscript and for the thoughtful and constructive comments and suggestions, which will help improve the overall quality of this manuscript. Our responses are denoted in **red**.

**General Comments:**

This manuscript presents a good showcase of the use of the next-generation global model MUSICAv0 with regional refinements to study the summertime distribution of ozone and its precursors in the Southeast Michigan region (SEMI) evaluated against the observations from the MOOSE campaign in 2021 and ground-based measurements. As one of the first studies to evaluate MUSICAv0 simulations with an extended campaign, this manuscript would be a notable publication. The study discusses in particular the effect on model performance of using higher grid resolution and implementing a diurnal cycle of anthropogenic NO emissions. It is shown that higher grid resolution is more important for simulating the distribution of O3 precursors than O3 itself, while implementing a diurnal cycle of anthropogenic NO emissions can improve model performance for nighttime O3. This conclusion is in agreement with the other modelling studies.

While this study clearly shows the advantage of using a global model with regional refinements, such as MUSICA, over the conventional global model, the manuscript does not discuss how these new generation models might improve on regional models. Perhaps the authors can add a short discussion on this issue and how these new generation models can be applied to better study regional air quality problems.

I recommend that this manuscript be published with the following comments and suggestions:

We thank Reviewer #2 for their careful observations, and appreciate their feedback and recommendations for improving the manuscript. We have carefully gone through all of your comments, and have addressed them below and in the main text.

**Section 1**

**Line 56:** A brief description of the instruments involved in the MOOSE campaign could be included here to provide a more comprehensive introduction to the campaign.

Response: To show the scope of MOOSE, we have added the sentence: "The MOOSE observations included a mobile lab with detailed measurements of ozone and its precursors, ground-based remote sensors (i.e., Pandora), and an airborne remote sensor (i.e., GCAS)."

**Line 100:** Please explicitly mention CAMS-GLOB-ANTv5.1 here, as there are a number of CAMS emission datasets. The resolution of the emission data (0.1 degree, ~10 km) can also be mentioned here to illustrate that the emission resolution is comparable to the model grid resolution. Please add a reference to the CAMS emissions dataset used here.

Response: The details on the emission datasets used for the simulations are presented in Section 2.1.3. As this sentence is about the diurnal variation, mention of the specific anthropogenic emissions has been left out here.

**Line 103:** Please add "emissions" after the end of the sentence

Response: The correction has been made.

**Section 2.1.2**

- Can the authors explain why the ne30x8 configuration covers the entire CONUS instead of just over Michigan?

Response: The ne30x8 configuration over CONUS is the default resolution used in MUSICAv0 and is mentioned in lines 110-112 (https://wiki.ucar.edu/spaces/MUSICA/pages/418448638/MUSICA+Home). The authors decided to use this grid mesh rather than create a new grid mesh over Michigan at 1/8-degree because it is a ready-to-use configuration with many of the input datasets readily available and in an NCAR repository. It was also a way of gauging the efforts and computational cost associated with creating a new grid mesh versus using an already available mesh.

**Line 135:** Can you include a reference to the Community Mesh Generation Toolkit?

Response: A citation for this software has been added.

**Section 2.1.3**

-    A more updated version of the CAMS-GLOB-ANT dataset should be considered in the future.

For temporal profiles, the CAMS-GLOB-TEMPO datasets may be useful.

Response: At the start of this work, the latest available inventory was used. A note of this to be considered in the future has been added in the conclusions section as: "Future work should also take into consideration the use of a more updated version of the CAMS-GLOB-ANT emissions, as well as the diurnal variation profiles of CAMS-GLOB-TEMPO (Guevara et al., 2021; Soulie et al., 2024), or more regional emission inventories such as the National Emission Inventory (NEI) from the US EPA."

**Section 2.1.4**

-    Apart from the diurnal cycle of NOx emissions, the evolution of the PBL probably plays a role in the daytime and nighttime O3 and NOx concentrations. Can the authors comment briefly on this?

Response: We acknowledge that apart from the diurnal cycle for anthropogenic NO emissions, the evolution of the planetary boundary layer (PBL) can play a significant role in O3 and NOx formation. We have added this statement in the conclusions acknowledging this uncertainty: "In addition, we acknowledge that apart from applying a diurnal cycle for anthropogenic NO emissions, the evolution of the PBL can also play a significant role in the formation of $O_3$ and $NO_x$. In the daytime, a rising PBL can mix surface $NO_x$ and VOCs upwards, reducing $O_3$ concentrations near the surface, while in the nighttime, a shallower PBL can trap emissions near the surface leading to higher $NO_x$ titration. Uncertainties associated with the PBL could lead to underpredictions of $NO_2$ in the model and misrepresentations of

$O_3$ peaks."

**Section 2.2.1**

**Table 2:** First column, first row: "Selected" VOCs

Response: This has been corrected in Table 2.

**Section 3.1**

- The diurnal cycle of NO emissions probably plays little role in meteorology. Please consider focusing the discussion on the effect of model grid resolution and select specific time periods where the simulations of the two resolutions show significant discrepancy for discussion.

Response: We have expanded the discussion here and have elaborated a bit further on the meteorological consistencies and inconsistencies. We have also included two additional time series (see Figure 3) plots for wind speed and wind direction to further show the meteorological performance from simulation to simulation.

Expanded Section 3.1: "SEMI is a region that faces unique air quality challenges due to large industrial and automotive activity, dense population, and geographic factors. SEMI has a diverse terrain, ranging from highly urbanized areas, such as the city of Detroit, expansive agricultural lands in more remote areas, and forests surrounded by both inland and coastal lakes. The region consists of a relatively flat terrain, with a humid continental climate. Additionally, large air masses of humidity can be transported into the region from the Great Lakes (i.e., Lake Huron and Lake Erie) through the lake effect winds (Scott and Huff, 1996). A time series along the AML track of meteorological values – temperature, relative humidity, planetary boundary layer height, cloud total, wind speed, and wind direction – from the models (and observations for temperature and relative humidity) are shown in **Fig. 3**. During the campaign period in the summer of 2021, temperatures reached up to approximately 305 K and relative humidity to almost 100%. The planetary boundary layer reached more than 2500 m on most days, while cloud total was relatively varied. Modeled wind speeds follow the trend for the campaign period quite well, but are comparatively high compared to the observations, while wind directions perform generally well except on some specific days. The AML track covered a large part of the SEMI region, making its way through both very urban and rural areas. Meteorological parameters, such as temperature, are highly impacted by urbanization through the reductions in vegetated land cover and increases in energy consumption (Wang et al., 2021). Urbanization can lead to higher temperatures, and thus increasing $O_3$ production. In the simulations presented here, meteorological parameters (i.e., temperature and horizontal winds) are nudged towards reanalysis data to obtain a more realistic depiction of reality in the coarser resolution regions, leaving the regional refinement area to freely run, as the resolution of the refined area is finer than the resolution of the reanalysis dataset that is being used. Regional refinement grids, with high horizontal resolution, are capable of resolving areas with large geographical differences (Jo et al., 2023). Meteorological fields in these simulations are generally consistent indicating that meteorology is performing similarly, even with the changes in horizontal resolution. Although temperatures, relative humidity, and planetary boundary layer height remain consistent among all the simulations, cloud total varies between the simulations, which can significantly impact photochemical production."

[Figure]

**Figure 3: Time series of (a) temperature, (b) relative humidity, (c) planetary boundary layer height, (d) cloud total, (e) wind speed, and (f) wind direction along the Aerodyne Mobile Laboratory (AML) track. Measurements of temperature and relative humidity were available and displayed as black x's in Fig. 3a and 3b. The model results are shown in red (ne30x8) and blue (ne30x16) corresponding to horizontal resolutions. The dashed lines represent model simulation results when adding the diurnal cycle for nitric oxide anthropogenic emissions, color-coded to their respective horizontal resolution.**

**Section 3.2**

**Line 326:** Please state in the text that Fig. 4 is a time series of hourly averaged diurnal profiles of ozone concentrations over a specific time period.

Response: As the reviewer suggests, text further describing Fig. 4 has been added. Here is the updated figure description: "The evaluation of the four model simulations with stationary measurements for $O_3$ at seven locations in SEMI – Allen Park (Suburban Downwind), Detroit-E 7 Mile (Suburban), New Haven (Rural), Oak Park (Suburban, Near Highway), Port Huron (Urban Port), Warren (Suburban), and Ypsilanti (Suburban, Near Highway) – are shown in **Fig. 4** as a time series of their hourly averaged diurnal profiles during the MOOSE campaign."

**Figures 4 and 5:** Please try to show time series of O3 and NO2 concentrations from the same stations in the same order for better comparison

Response: O3 and NO2 are not available at all sites, so Figures 4 and 5 necessarily show different sites. Out of the shown sites, only one – Detroit – E 7 Mile – has data available for both O3 and NO2. We have included all of the sites in SEMI with data available to show comparison with a wide range of sites in different locations throughout the area. Additionally, in the discussion section (Sec. 4), we include diurnally averaged plots (Figs. 15 and 18) that compare multiple species at several locations to have a more representative perspective on O3 production and loss in the area of study.

**Line 358:** Can the authors explain in more detail how O3 concentrations are affected by the aforementioned effect on NO2 concentrations?

Response: We have included a more detailed explanation of how O3, NOx, and VOCs are intertwined in the beginning of Section 3, to tie together the different evaluations being done, as so: "$O_3$ concentrations are highly associated with $NO_2$, where $NO_x$, in general, plays a critical role in the photochemical production and destruction of $O_3$ in the presence of sunlight. $O_3$ production in the troposphere is largely dependent on the availability of $NO_x$ and VOCs, and can give great insight on $O_3$ control. This dependency is classified into $NO_x$- and VOC-limited regimes. In a $NO_x$-limited regime, the rate of $O_3$ production relies on the abundance of $NO_x$ and increases with $NO_x$ concentrations, but is not dependent on the concentrations of VOCs (Wang et al., 2019) . In action, decreasing $NO_x$

concentrations would lead to reductions on $O_3$ (Jacob, 1999). On the other hand, in a VOC-limited regime (or $NO_x$-saturated regime) the rate of $O_3$ production increases with VOC concentrations and is not dependent of $NO_x$ (Wang et al., 2019), therefore reducing the amount of VOCs would lead to reductions in $O_3$ (Jacob, 1999). The chemical relationship between $O_3$-$NO_x$-VOCs is critically important for defining mitigation strategies set to improve $O_3$ from region to region."

**Section 3.3**

-   Can the authors also discuss whether there are significant differences between daytime and nighttime concentrations between the four simulations?

Response: Overall differences between daytime and nighttime concentrations between the simulations are detailed in Sec. 3.2 of the main text, where the simulations are compared to stationary measurements of O3 and NO2. In Sec. 3.3, we omit a discussion on daytime and nighttime concentrations because we are summarizing the data from AML and the models with a Taylor diagram. We do this because for data along the AML track, the mobile lab was moving on different days in different locations across SEMI

(some urban areas, some more remote areas, etc.), and stationary at night. Comparing hourly averaged data for this section, may have been misleading if trying to compare the entirety of the campaign. If we were to include specific case study days, a discussion of the daytime and nighttime performance would have been critical. The goal of using the Taylor diagram here was to summarize the overall model performance compared to these observations.

**Section 3.4**

-   The authors should better illustrate how the Pandora measurements can be related to the stations and AML measurements and how these comparisons can lead to the different performance of simulated O3 concentrations.

Response: This is a great suggestion, as we realize we do not quite explain why we use different datasets for the model evaluation. We have elaborated further at the beginning of the results section (Sec. 3) to emphasize why model evaluations are needed – "We evaluate the models using diverse datasets from MOOSE for a comprehensive analysis, as no single dataset has the ability to capture all aspects of atmospheric composition (e.g., emissions, chemistry, transport, meteorology). These different datasets can also help capture different aspects of a model such as near-surface chemistry (i.e., in-situ measurements) and column burdens (i.e., aircraft-based remote sensing), to determine model skill, characterize model errors, improve model representation, and measure our confidence in the model results for reproducing reality" –  and what needs to be considered, as well as adding a comment at the beginning of Section 3.4: "Both Pandora monitoring sites (SWDetroitMI and DearbornMI) were located in an industrial and high-traffic setting, providing continuous observations in urban conditions and complementing the other observations".

**Section 3.5**

- Instead of narrative in the text, the authors could include wind vectors or maps of meteorological variables to illustrate how the different models capture the NO2 plumes at the different times shown, as the readers may not be familiar with the geographical locations shown.

Response: We have added maps for each of the days to include wind vectors and temperature to the supplemental information (Figures S16-S31), and have referenced them in the main text (line 534) as so "The direction of the pollution plume  are supported by plots of temperature and wind vectors in Figs. S16-S31 in the SI for each of the flight days".

**Section 4**

**Figures 13 and 14:** The authors should explain why they show the conservatively regridded model outputs in panel (c).

Response: We thank the reviewer's attention to these figures. We have added an introduction to this discussion to motivate the comparison of the regridded high resolution output to the coarser resolution:

"To quantitatively assess the impact of the finer resolution on the simulation of ozone and its precursors, the ne30x16 (7 km) results have been conservatively regridded to the ne30x8 grid.  These regridded results illustrate the impact model resolution can have on atmospheric chemistry."

**Line 512:** The authors should explain the consequences of the regridding method not being able to reproduce the higher resolution simulation results

Response: We've incorporated a couple of sentences in/around line 589 to address these consequences:

"When the model is run at 1/16° horizontal resolution, localized features (e.g., pollution plumes, sharp emission gradients) are better resolved and land use is better represented."

**Figure 15:** Please consider also including the observed concentrations of O3, NO and NO2 in the time series to better illustrate which model configuration is closer to the observed values.

Response: Thank you for your suggestion. The available observed values for O3, NO, and NO2 at these stationary monitoring locations are presented in Section 3.2 in Figures 4 and 5.

**Section 5**

**Line 618:** In addition to the South Korean simulation, can the authors compare their work with other studies using MUSICA over other regions of the USA?

Response: We have included a sentence comparing this work with Schwantes et al. (2022) which used

MUSICAv0 to study the Southeastern US: "Schwantes et al. (2022) found that $O_3$ was better simulated over urban regions across the Southeastern US, especially when using a ~14 km regional refinement grid and updated chemistry in MUSICAv0. This work took into consideration a finer grid resolution mesh (~7 km) and compared to ~14 km to show that regional refinement improves $O_3$

representativeness in the model."

-    The authors should consider briefly discussing how MUSICA or similar next-generation global models with regional refinement capability can be used to formulate regional/local air pollution monitoring and mitigation strategies.

Response: We have expanded Sec. 4 to include a better link with Sec. 3. We have also established a better discussion on different regimes governing SEMI and what potential mitigation strategies could be applied to the area to improve air quality. We have also discussed future work in helping define potential mitigation effects.

These linkages are reflected in Section 4 as so:

- "This difference results in an improvement for the ne30x16 simulations based on the findings from in Fig. 4, where peak $O_3$ performed best in the finer resolution simulations when compared to the surface sites."

- "These findings are directly supported by **Fig. 3**, where although temperatures between the simulations are not significantly different, there are changes in cloud totals and winds that could impact solar radiation and thus the isoprene emissions. The differences in temperature between the resolutions are also illustrated in the maps in **Fig. S16-S31** in the SI."

An extended discussion on the different regimes governing SEMI and potential mitigation is added to the end of Section 4: "The findings of this study show that $O_3$ production in SEMI is strongly governed by the spatial distribution of emissions and different chemical regimes. The urban location analysis showed that Detroit, which is a major industrial hub in the region, is consistent with a VOC-limited regime, where in the daytime, $O_3$ concentrations are suppressed by high $NO_x$ titration, but can become sensitive to changes in VOCs during peak $O_3$ times. The suburban and remote location analysis (i.e.,

Allen Park and New Haven, respectively) showed that they are in a more $NO_x$-limited regime, where higher BVOCs and lower $NO_x$ titration can lead to more efficient $O_3$ production. The spatial distribution is seen more clearly as we move towards finer resolutions indicating more realistic emissions.

In VOC-limited regimes, targeting reductions in VOCs, such as those from the industrial sectors, is crucial compared to reductions in $NO_x$, as it could lead to temporary increases in $O_3$ production. In

$NO_x$-limited regimes, where $NO_2$ drives $O_3$ production, reductions in transportation emissions and long- range transport would decrease $O_3$. The improvement in model representation of $NO_2$ and in turn, $O_3$, during rush hour times (**Fig. 4-5**) shows how emissions can be misrepresented in the models. It is necessary that future work considers incorporating higher resolution temporal profile and regional emissions to better distinguish different $O_3$ processes. Future work should also explore the impacts of targeting the contribution of different emission scenarios in SEMI to demonstrate the impact of different regulatory decision-making. ”

---

## Author Response (AR2)

**Response to Correction from Topic Editor**

Dear Dr. O'Connor,

Thank you very much for your positive recommendation for overseeing the review process of our manuscript. We greatly appreciate your thoughtful guidance and the constructive feedback we received from the reviewers.

Regarding the comment that arose from the validation step, we have revised the reference list to remove any citations listed as "in preparation" that are not yet publicly available. We have uploaded the updated version of the manuscript and all supporting materials.

Thank you again for your time and support throughout this review process.

Best Regards,

Noribeth Mariscal (on behalf of all co-authors)